# Rademacher expansion of modular integrals

**Marco Maria Baccianti[1]★, Jeevan Chandra[2]†, Lorenz Eberhardt[1]‡,**
**Thomas Hartman[2]° and Sebastian Mizera[3,4,5]§**

**1** Institute for Theoretical Physics, University of Amsterdam, Amsterdam, 1098XH, NL
**2** Department of Physics, Cornell University, Ithaca, NY 14853, USA
**3** Department of Physics, Princeton University, Princeton, NJ 08544, USA
**4** Princeton Center for Theoretical Science, Princeton University, Princeton, NJ 08544, USA
**5** Institute for Advanced Study, Princeton, NJ 08540, USA

★ m.m.baccianti@uva.nl ,   † jn539@cornell.edu ,   ‡ l.eberhardt@uva.nl ,
° hartman@cornell.edu ,   § smizera@ias.edu

## Abstract

We develop a method to evaluate integrals of non-holomorphic modular functions over the fundamental domain of the torus with modular parameter $\tau$ analytically. It proceeds in two steps: first the integral is transformed to a Lorentzian contour by the same strategy that leads to the Lorentzian inversion formula in CFT, and then we apply a two-dimensional version of the Rademacher expansion. This computes the integral in terms of an expansion sensitive to the singular behaviour of the integrand near all the Lorentzian cusps $\tau \to i\infty$, $\bar{\tau} \to x \in \mathbb{Q}$. We apply this technique to a variety of examples such as the evaluation of string one-loop partition functions, where it leads to the first analytic formula for the cosmological constants of the bosonic string and the $SO(16) \times SO(16)$ string.

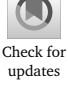

# 1 Introduction

In theoretical physics one often encounters quantities that can be reduced to integrals over the moduli space of complex structures of a torus. This moduli space can be realized as the famous keyhole region $\mathcal{F}$ in the $\tau$-plane known as the *fundamental domain*, see Figure 1. Examples occur prominently in computations of closed string theory amplitudes at one-loop level, where the integral over the modular parameter of the torus is of this sort [1–4]. Other examples include Donaldson-Witten theory, see e.g. [5] or threshold corrections to supergravity, see e.g. [6,7].

Such integrals also appear in Euclidean inversion formulas of 2d CFTs. In analogy with the higher dimensional case, one can decompose partition functions of 2d CFTs into a preferred basis, such as the eigenbasis of the Laplacian on the fundamental domain which loosely play the role of conformal partial waves. This spectral viewpoint on 2d CFT has recently been investigated [8–13].

The pillow coordinates [14, 15] $z = \lambda(\tau) = \frac{\vartheta_2(\tau)^4}{\vartheta_3(\tau)^4}$ map the moduli space of the four-punctured sphere to (six copies of) the moduli space of the once-punctured torus, and both string amplitudes for the four-punctured sphere as well as inversion formulas for 2d CFTs for the four-point function involve the same kind of integrals.

Except in special circumstances which usually involve the 'unfolding' of the integral from the fundamental domain to a vertical strip $-\frac{1}{2} \leq \operatorname{Re}\tau \leq \frac{1}{2}$ such as in the Rankin-Selberg method [16], these integrals can only be evaluated numerically and it is difficult to get a better handle on them.

**Lorentzian inversion.** In the inversion formula of higher-dimensional CFTs, the *Lorentzian* inversion formula [17,18] gave some amount of analytic control over the analogous integral. In the present case, such a Lorentzian inversion formula involves a two-dimensional contour deformation of the integral over the fundamental domain in the *complexification* of the moduli space of tori. Concretely, this means that one promotes $\tilde{\tau} := -\bar{\tau}$ to an independent variable in the upper half plane $\mathbb{H}$ and views the integrand as a modular function of $(\tau, \tilde{\tau}) \in \mathbb{H} \times \mathbb{H}$. We assume that the integrand is analytic in two variables on $\mathbb{H} \times \mathbb{H}$. The original integration contour $\mathcal{F} \subset (\mathbb{H} \times \mathbb{H})/\operatorname{SL}(2,\mathbb{Z})$ can now be freely deformed in this complexified space. For a Lorentzian inversion formula, the contour would be holomorphically split, meaning that it is a product contour in $\tau$ and $\tilde{\tau}$. This is our first result. Taking the integration measure to be $\mathrm{d}^2\tau = \mathrm{d}\operatorname{Re}\tau \wedge \mathrm{d}\operatorname{Im}\tau = \frac{1}{2i}\mathrm{d}\tau \wedge \mathrm{d}\tilde{\tau}$, we will show that the integral of a modular-invariant function over the fundamental domain has the Lorentzian expression[1]

$$\int_{\mathcal{F}} \mathrm{d}^2\tau\, f(\tau,\tilde{\tau}) = \frac{1}{12i}\int_0^{i\infty}\mathrm{d}\tau \int_0^{i\infty}\mathrm{d}\tilde{\tau}\,\operatorname{Disc} f(\tau,\tilde{\tau}) = \frac{1}{12i}\int_0^{i\infty}\mathrm{d}\tau \int_{-1}^{1}\mathrm{d}\tilde{\tau}\, f(\tau,\tilde{\tau}), \quad (1)$$

where $\operatorname{Disc} f = f(\tau,\tilde{\tau}-1) - f(\tau,\tilde{\tau}+1)$. In the last expression, the $\tau$-contour connects $0$ and $i\infty$ in the upper half-plane and the $\tilde{\tau}$ contour connects $-1$ and $1$ in the upper half plane. There are actually several small variations on this formula, which we mention in section 2.6.2. It is for example straightforward to check (1) for the upper half-plane measure $f(\tau,\tilde{\tau}) = \frac{1}{(\operatorname{Im}\tau)^2} = \left(\frac{2i}{\tau+\tilde{\tau}}\right)^2$, but also more complicated examples can be checked numerically. The derivation is a close analogue of the derivation of the Lorentzian inversion formula [17,18].

---

[1]We also assumed that the integrand is parity symmetric, i.e. $f(\tau,\tilde{\tau}) = f(\tilde{\tau},\tau)$. If it is not, we can just replace it by $\frac{1}{2}(f(\tau,\tilde{\tau}) + f(\tilde{\tau},\tau))$ and apply the formula. In a similar way, the formula can be applied to modular functions for finite-index subgroups of $\operatorname{SL}(2,\mathbb{Z})$ by summing over images first.

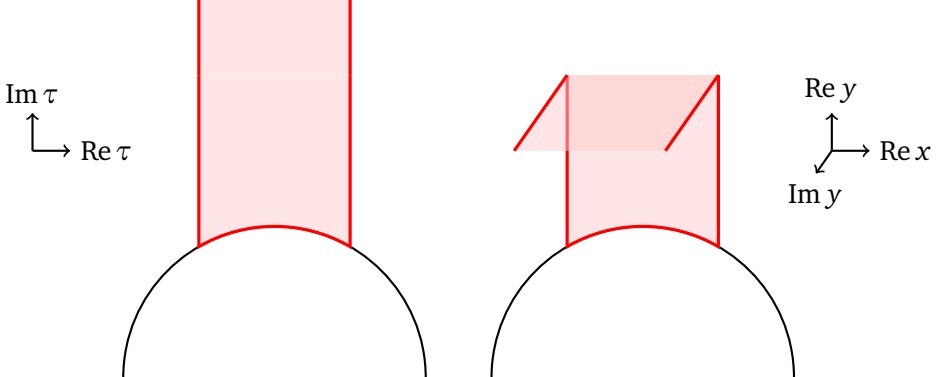

Figure 1: The integration contour over the original (left) and modified (right) fundamental domain. On the left, we have $(\operatorname{Re} x, \operatorname{Re} y) = (\operatorname{Re} \tau, \operatorname{Im} \tau)$.

**Behaviour at the cusps.** In most cases of interest, the integrals over the fundamental domain are usually only marginally convergent or need regularization of some kind, which reflects the presence of IR-singularities in the string amplitude case[2] or the negative Casimir energy on the torus in the case of the CFT partition function. The presence of this divergence is a double-edged sword. The necessity of a regularization significantly complicates the precise definition of these integrals and makes direct numerical integration much harder. Indeed, a natural way to define the integral over the fundamental domain is suggested by string theory. In the context of the one-loop partition function, the worldsheet degenerates to a very long torus in the dangerous cusp region $\tau \to i\infty$. QFT tells us that $\operatorname{Im} \tau$ is identified with a Euclidean Schwinger parameter of the low-energy field theory and should be Wick-rotated to a Lorentzian Schwinger parameter. The resulting contour is depicted in Figure 1. This means that already the starting point on the LHS of (1) is naturally defined through a contour that becomes Lorentzian near the cusp $\tau \to i\infty$. One has two choices depending on the direction in which we Wick rotate. In the string theory context, the $i\varepsilon$ prescription tells us to turn left near the cusp [21, 22]. We call the corresponding modified contour $\mathcal{F}_{i\varepsilon}$ and the contour turning in the other direction $\mathcal{F}_{-i\varepsilon}$. If the original integrand $f(\tau, \tilde{\tau} = -\bar{\tau})$ was real before we complexified, after this modification it is no longer real. However, the imaginary part is simple since it only originates from the region near the cusp and we are thus not really interested in it. We will define a regulated integral over the fundamental domain $\mathcal{F}$ by taking the average of both choices of Wick rotations. We will denote this by a slashed integral $\fint_{\mathcal{F}} = \frac{1}{2} \left( \int_{\mathcal{F}_{i\varepsilon}} + \int_{\mathcal{F}_{-i\varepsilon}} \right)$ in analogy with the principal value prescription for one-dimensional integrals. The identity (1) still holds with these modifications, provided that the contour on the RHS approaches the 'Lorentzian cusps' in the right direction. The appropriate contour is depicted in Figure 4.

**Rademacher formula.** Despite these complications, the presence of divergences near the cusps actually opens up an avenue to *analytically* evaluate these integrals. The results we find are expressed as an infinite sum over all the *Lorentzian* cusps of the integrand. These are labelled by rational numbers $0 \leqslant \frac{a}{c} < 1$ with $\frac{a}{c} \in \mathbb{Q}$ and represent the limit $\tau \to i\infty$ and $\tilde{\tau} \to \frac{a}{c}$. Under a certain growth condition, we show that the Lorentzian formula (1) can be further manipulated as follows:

$$\fint_{\mathcal{F}} \mathrm{d}^2\tau \, f(\tau, \tilde{\tau}) = \sum_{c=1}^{\infty} \sum_{\substack{a=0 \\ (a,c)=1}}^{c-1} \int_{\longrightarrow} \mathrm{d}\tau \int_{C_{a/c}} \mathrm{d}\tilde{\tau} \left[ \frac{1}{12i} \left( \tau - \tilde{\tau} + \frac{2a}{c} \right) + i s(a,c) \right] f(\tau, \tilde{\tau}). \quad (2)$$

---

[2]There are some rare string theoretic quantities that are free from such IR-divergences such as the one-loop cosmological constant in non-supersymmetric string theories [19, 20].

The integral on the RHS is still holomorphically split with $\tau$ running over an unbounded horizontal contour in the upper half-plane and $\tilde{\tau}$ around the Ford circle $C_{a/c}$ in a clockwise sense. These are circles in the upper half-plane that touch the real axis at the fraction $\frac{a}{c}$. Even though this is a homologously trivial contour, the integral is finite and doesn't vanish because the integrand has an essential singularity at $\frac{a}{c}$. The quantity $s(a, c)$ appearing on the RHS of (2) is the Dedekind sum that also appears in the transformation law of the Dedekind eta function. We will give a precise definition in (49). Dedekind sums are very well-studied by analytic number theorists and can be evaluated efficiently thanks to their reciprocity relation (E.19).

Even though (2) may look daunting and much more complicated than what we started with, the main point is that the integrals on the RHS are simple to evaluate analytically. Indeed, they are only sensitive to the local behaviour of the integrand near the Lorentzian cusp since we can push the $\tau$ contour to high imaginary parts and contract the circle $C_{a/c}$. The singular pieces that control the integral on the RHS of (2) are the analogue of the polar terms of weakly holomorphic modular functions. Just as in the holomorphic setting, knowledge of all the polar contributions thus completely determines the integral! For comparison, the Lorentzian inversion formula for CFT correlators [17,18] has contributions near the lightcone singularity that capture the high-spin asymptotics of the CFT data [17,23,24]. The sum (2) goes a step further by unwrapping the integral onto the infinite set of lightcone singularities of twist operators on the Lorentzian cylinder, giving an exact evaluation of the integral.

The expansion of the integral over the fundamental domain in terms of its Lorentzian cusps is very similar to the Rademacher expansion that applies for contour integrals of holomorphic modular objects. In fact, the standard Rademacher expansion forms an important step in the derivation of (2). For this reason, we refer to (2) as a two-dimensional Rademacher contour. The Rademacher expansion is a version of the Hardy-Littlewood circle method and has found many applications in theoretical physics. It shows up in the matching of the supersymmmetric index with the gravitational path integral [25,26], and was also generalized to Siegel modular forms in this context [27,28]. It was also recently effectively applied to the computation of one-loop open-string amplitudes [3,29] and we plan to apply the techniques presented in this paper in the future to a similar evaluation of the closed string one-loop amplitude. The Rademacher method was also applied in the context of pure quantum gravity in $AdS_3$ [30] to try to get a better handle on the Maloney-Witten-Keller partition function [31,32] and we hope that the techniques presented in this paper will be useful in this setting as well.

In the holomorphic setting, the Rademacher contour converges for *negative* weights (although this may sometimes be relaxed [33,34]). The analogue of this condition in the non-holomorphic setting can be formulated as a growth condition on the integrand when restricted to the contour $\tau \in \longrightarrow$ and $\tilde{\tau} \in C_{a/c}$, see eq. (B.1) for the precise formula. It ensures the convergence of the sum over $c$ on the RHS of (2).

**Example.** As a simple example, when we apply (2) to the bosonic string partition function $f(\tau, \tilde{\tau}) = \frac{1}{(\mathrm{Im}\,\tau)^{14}|\eta(\tau)^{24}|^2}$, we obtain

$$Z = \frac{(4\pi)^{15}}{24 \cdot 13!} \sum_{c=1}^{\infty} \sum_{\substack{a=0 \\ (a,c)=1}}^{c-1} \frac{e^{2\pi i \frac{a+a^*}{c}}}{c^2} \left[ 12 i c\, s(a,c) J_{13}\left(\tfrac{4\pi}{c}\right) + J_{12}\left(\tfrac{4\pi}{c}\right) - J_{14}\left(\tfrac{4\pi}{c}\right) \right] + \frac{(4\pi)^{14} i}{4 \cdot 13!}. \quad (3)$$

Here, $J_\nu(x)$ is the Bessel function of the first kind that appears frequently in the relevant integral over the Lorentzian cusps. The second term is the imaginary part that can be evaluated analytically as we promised above. Even though the partition function $Z$ appears in every first course on string theory (where it is often incorrectly stated to not make sense because of the divergence from the tachyon), this is the first closed-form analytic expression for it that we are aware of. Notice that the sum on the RHS is very much dominated by the first few values

of $c$ and it already suffices to keep the first few terms to get an accurate approximation to the integral. For convenience, we give example implementations of the Rademacher formula in a Mathematica notebook attached as an ancillary file to this submission.

**Outline.** This paper is organized as follows. We begin in section 2 to explain the precise setup of the modular integrals more carefully. We then explain the contour deformation leading to the holomorphically split formula (1). From there, we apply several further contour manipulations and eventually derive the Rademacher formula (2). Even though we have decided to keep the discussion somewhat informal, our derivation is completely rigorous and some of the more technical steps are relegated to Appendices A and B. In section 3, we discuss several applications of the main formula to integrals of interest, such as the bosonic string one-loop partition function mentioned above, the one-loop mass-shift of the massive string state in type II strings, the one-loop cosmological constant of type 0 string theory and integrals of rational CFT partition functions.

## 2 Contourology

In this section, we will derive the two formulas eq. (1) and (2). Let us give an outline of the involved steps:

1. We first discuss the proper contour that we use to compute the integral.

2. This two-dimensional contour in $(\tau, \tilde{\tau})$ is then holomorphically split into a contour integral over $\tau$ and over $\tilde{\tau}$. For this, we map the integral to an integral over a cross ratio via the pillow coordinate map. The deformation is then simple in the cross ratio space and can then be translated back to the modular parameter.

3. The holomorphically split contour is further deformed and manipulated. This leads to two different terms in eq. (37). It may not be immediately obvious why this step is necessary and we comment on this in section 2.6.1.

4. The Rademacher expansion is applied. This is rather standard for one of the terms and leads to the terms linear in $\tau$ and $\tilde{\tau}$ in (2). We review the necessary background on the Rademacher expansion. The second term is also expanded into a Rademacher contour in both $\tau$ and $\tilde{\tau}$. This expansion can be reduced back to a single sum over Ford circles, but with multiplicities. These multiplicities are counted by a function $\mu(\frac{a}{c})$ obeying a certain recursion relation (48).

5. Finally, we relate $\mu$ to the Dedekind sum that appears in (2). This step is entirely number theoretic.

We numbered the subsections according to these steps. In section 2.6, we make some further remarks on the derivation.

### 2.1 Step 1: Setting up the contour

We will first complexify the integrand. We set $\tilde{\tau} = -\bar{\tau}$ and consider $\tau$ and $\tilde{\tau}$ to be independent. Both $\tau$ and $\tilde{\tau}$ are constrained to the upper half plane. We now carefully define the integral

$$I = \oint_{\mathcal{F}} \alpha(\tau, \tilde{\tau}). \tag{4}$$

Notice also that $d^2\tau = \frac{1}{2i}d\tau \wedge d\tilde{\tau}$. It is convenient to use this holomorphically factorized measure. In the introduction we expressed the formulas in terms of $f$,

$$\alpha(\tau, \tilde{\tau}) = \frac{1}{2i} f(\tau, \tilde{\tau}) d\tau \wedge d\tilde{\tau}, \tag{5}$$

but it is more convenient in the following to consider also the differentials as part of the integrand. The form $\alpha(\tau, \tilde{\tau})$ is invariant under joint modular transformations acting as follows,

$$\alpha\left(\frac{a\tau + b}{c\tau + d}, \frac{a\tilde{\tau} - b}{-c\tilde{\tau} + d}\right) = \alpha(\tau, \tilde{\tau}). \tag{6}$$

Note the minus signs in the second argument, which we can write as

$$-\frac{a(-\tilde{\tau}) + b}{c(-\tilde{\tau}) + d}, \tag{7}$$

arising from putting $\bar{\tau} = -\tilde{\tau}$. This implies that $f$ transforms covariantly under joint modular transformations with weight $(2, 2)$ in $\tau$ and $\tilde{\tau}$. We also make the two assumptions

$$\alpha(\tau, \tilde{\tau}) = -\alpha(\tilde{\tau}, \tau), \qquad \alpha(\tau, \tilde{\tau})^* = -\alpha(-\bar{\tau}, -\bar{\tilde{\tau}}). \tag{8}$$

The minus sign in the first equation only comes from the differential forms and these conditions are equivalent in terms of $f$ as in (5) to

$$f(\tau, \tilde{\tau}) = f(\tilde{\tau}, \tau), \qquad f(\tau, \tilde{\tau})^* = f(-\bar{\tau}, -\bar{\tilde{\tau}}), \tag{9}$$

with $f$ as in (5). The first condition means that the integrand is parity symmetric, while the second means that it is real when restricted to the real slice $\tilde{\tau} = -\bar{\tau}$. None of them are essential, but it is convenient to assume for the derivation.

The integration contour defining the integral $\fint$ was already described in words in the introduction, but let us be more precise. We integrate over the fundamental domain of the diagonal $\mathbb{H}/SL(2, \mathbb{Z}) \subset (\mathbb{H} \times \mathbb{H})/SL(2, \mathbb{Z})$ with a modification at the cusp. To make this precise, let for $L > 1$

$$\mathcal{F}_L = \left\{\tau, \tilde{\tau} \in \mathbb{H} \,|\, \bar{\tau} = -\tilde{\tau}, \,|\tau| > 1, \,-\tfrac{1}{2} \leqslant \operatorname{Re}\tau \leqslant \tfrac{1}{2}, \,\operatorname{Im}\tau \leqslant L\right\}. \tag{10}$$

We can then integrate over $\mathcal{F}_L$ as usual. Near the cusp, we define coordinates $y = \frac{\tau + \tilde{\tau}}{2i}$ and $x = \frac{\tau - \tilde{\tau}}{2}$. The contour is then modified by letting the $y$ contour turn into the complex plane, i.e. we define the contour over the complement of $\mathcal{F}_L$ as

$$\int_{\mathcal{F}\backslash\mathcal{F}_L} \alpha(\tau, \tilde{\tau}) = \int_{x \in \left[-\frac{1}{2}, \frac{1}{2}\right]} \int_{y \in [L, \infty)} \alpha(x, y) \to \int_{x \in \left[-\frac{1}{2}, \frac{1}{2}\right]} \int_{y \in [L, L - i\infty)} \alpha(x, y). \tag{11}$$

Here and from now on, we write $\alpha(x, y)$ etc. to denote $\alpha$ expressed in different coordinates. The choice of how to modify this contour is the counterpart of the Feynman $i\varepsilon$ prescription for the fundamental domain. It follows from matching it with the corresponding worldline prescription in the limit $\tau \to i\infty$ where the torus degenerates into a circle [21]. See [22, Section 2] for a more detailed discussion motivating this choice. We denote the deformed contour of the fundamental domain by $\mathcal{F}_{i\varepsilon}$, i.e.

$$\operatorname{Im}\int_{\mathcal{F}_{i\varepsilon}} \alpha(\tau, \tilde{\tau}) := \int_{\mathcal{F}_L} \alpha(\tau, \tilde{\tau}) + \int_{x \in \left[-\frac{1}{2}, \frac{1}{2}\right]} \int_{y \in [L, L - i\infty)} \alpha(x, y). \tag{12}$$

This contour is depicted in Figure 1. We denote the contour for which we turn in the other direction by $\mathcal{F}_{-i\varepsilon}$. The direction in which the contour turns does not matter for the real part and we have

$$\oint_{\mathcal{F}} \alpha(\tau,\tilde{\tau}) := \frac{1}{2}\int_{\mathcal{F}_{i\varepsilon}} \alpha(\tau,\tilde{\tau}) + \frac{1}{2}\int_{\mathcal{F}_{-i\varepsilon}} \alpha(\tau,\tilde{\tau}) = \text{Re}\int_{\mathcal{F}_{i\varepsilon}} \alpha(\tau,\tilde{\tau}). \tag{13}$$

This definition of the regularized integral generalizes in particular those discussed in the mathematical literature [35, 36] and the physics literature [6, 37]. We always modify the radial variable that parametrizes the distance near the cusp by the same deformation to turn left in the complex plane before reaching the cusp.

## 2.2 Step 2: Lorentzian integration formula

We will now derive the Lorentzian integration formula (1). We have included a small variation of this derivation in appendix C. We will for the moment not keep track of the modification of the contour near the cusp. We will reinstate it below eq. (23).

We first write the integral as an integral over the union of 6 different fundamental domains and divide by a factor of 6. The 6 fundamental domains make up a fundamental domain of the congruence subgroup

$$\Gamma(2) = \left\{ \begin{pmatrix} a & b \\ c & d \end{pmatrix} \middle| a \equiv d \equiv 1 \bmod 2, \ b \equiv c \equiv 0 \bmod 2 \right\}. \tag{14}$$

This is indeed an index 6 subgroup of $\text{SL}(2,\mathbb{Z})$ with the cosets to be of the form

$$\begin{pmatrix} 1 & 0 \\ 0 & 1 \end{pmatrix}, \quad \begin{pmatrix} 1 & 1 \\ 0 & 1 \end{pmatrix}, \quad \begin{pmatrix} 1 & 0 \\ 1 & 1 \end{pmatrix}, \quad \begin{pmatrix} 1 & 1 \\ 1 & 0 \end{pmatrix}, \quad \begin{pmatrix} 0 & 1 \\ 1 & 1 \end{pmatrix}, \quad \begin{pmatrix} 0 & 1 \\ 1 & 0 \end{pmatrix} \bmod 2. \tag{15}$$

The fundamental domain can be taken to be an ideal hyperbolic triangle with vertices 0 and 1 and $\infty$. It takes the form as depicted in Figure 2. In particular, $\Gamma(2)$ acts freely on the upper half plane.[3] This is not true for $\text{SL}(2,\mathbb{Z})$, since $i$ and $e^{\frac{2\pi i}{3}}$ are fixed points of $S$ and $ST$ respectively. This is the main reason why $\Gamma(2)$ is more convenient since we do not have to deal with such points.

Since $\mathbb{H}/\Gamma(2)$ has genus 0, we can find a bijective holomorphic map to the Riemann sphere, which we coordinatize by the cross ratio $z$. The map becomes unique if we fix three points. It is convenient to map 0 to 1, 1 to $\infty$ and $i\infty$ to 0. By composing with the quotient map $\mathbb{H} \to \mathbb{H}/\Gamma(2)$, we get the uniformizing map

$$\gamma : \mathbb{H} \longrightarrow \mathbb{CP}^1. \tag{16}$$

Such a map is called the Hauptmodul in the theory of modular forms. We now consider the inverse map

$$\gamma^{-1} : \mathbb{CP}^1 \longrightarrow \mathbb{H}. \tag{17}$$

This map is multivalued with monodromy around 0, 1 and $\infty$.[4] We can let the branch cuts run from 0 to $\infty$. Thus we can fix the principal branch by requiring that $\gamma^{-1}$ maps $\mathbb{C}\setminus[0,\infty)$ to the interior of the fundamental domain.

The monodromies are simple to determine since they correspond to the modular transformations that identify the boundaries of the fundamental domain. For the cusp $i\infty$, this

---

[3]As stated, this is not quite accurate since we considered $\text{SL}(2,\mathbb{Z})$ and every point is stabilized by $-\mathbb{1}$, corresponding to the $\mathbb{Z}_2$ automorphism of every torus. This is not relevant in the following.

[4]This is because the branch points are precisely those where several sheets meet. These points are the vertices of the hyperbolic triangle at $0,1,\infty$.

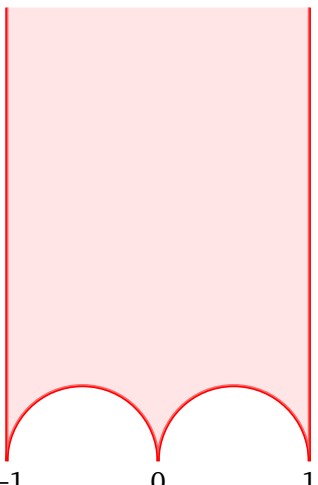

Figure 2: The fundamental domain of $\Gamma(2)$.

modular transformation is $T^2$, for the cusp at 0 it is $S^{-1}T^2S = ST^2S$, while for the cusp at 1 it is $(ST^{-1})^{-1}T^2(ST^{-1}) = TST^2ST^{-1}$. Since there are three monodromies with SL$(2,\mathbb{Z})$ monodromy matrices and hypergeometric functions have also three branch points with SL$(2,\mathbb{Z})$ monodromy matrices mixing the two linearly independent solutions of the hypergeometric differential equation, we can write the inverse map $\gamma^{-1}$ as the ratio of two hypergeometric functions.

Furthermore, we can improve this setup a little bit more by noting that the quotient group SL$(2,\mathbb{Z})/\Gamma(2) \cong S_3$[5] acts on the $z$-coordinate as $z \to 1-z$ and $z \to \frac{1}{z}$. This is clear since these are the unique invertible holomorphic maps on the Riemann sphere that permute the three cusps. Hence we can restrict our integration region to $|z| < 1$ since the contribution from $|z| > 1$ is identical because the integrand is invariant under all of SL$(2,\mathbb{Z})$.

We can thus write the integral as an integral over the unit disk $\mathbb{D}$,

$$I = \frac{1}{3}\int_{\mathbb{D}\backslash[0,1]} \alpha(z,\tilde{z}). \tag{18}$$

We now change to polar coordinates. Write $z = uv$, $\tilde{z} = uv^{-1}$ with $|v| = 1$ with $0 < u < 1$ and with $v \neq 1$ to exclude the branch cut. Thus we have

$$I = \frac{1}{3}\int_{u\in[0,1]}\int_{|v|=1} \alpha(u,v). \tag{19}$$

We can now deform the $v$ contour to wrap around the cuts only. Thus we get

$$I = \frac{1}{3}\int_{u\in[0,1]}\int_{v\in\mathcal{H}_\delta} \alpha(u,v), \tag{20}$$

where the Hankel contour $\mathcal{H}_\delta$ starts at $1+i\delta$ and goes around the branch cut and goes back to $1-i\delta$ with $\delta$ infinitesimal, see Figure 3.

Let us translate this back to $z$. Neglecting momentarily the small circle around the origin in $\mathcal{H}_\delta$, this gives

$$I = \frac{1}{3}\left(-\int_{z\in[i\delta,1+i\delta]}\int_{\substack{\tilde{z}\in[1-i\delta,\infty-i\delta)\\ z\tilde{z}<1}} + \int_{z\in[-i\delta,1-i\delta]}\int_{\substack{\tilde{z}\in[1+i\delta,\infty+i\delta)\\ z\tilde{z}<1}}\right)\alpha(z,\tilde{z}). \tag{21}$$

---

[5]$\Gamma(2)$ is a normal subgroup since it is the kernel of the mod 2 reduction.

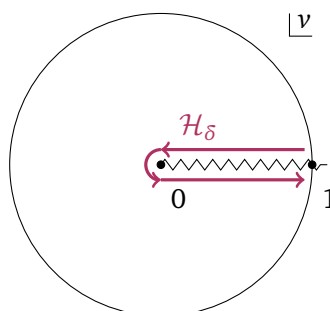

Figure 3: The Hankel contour $\mathcal{H}_\delta$ in the $\nu$-plane.

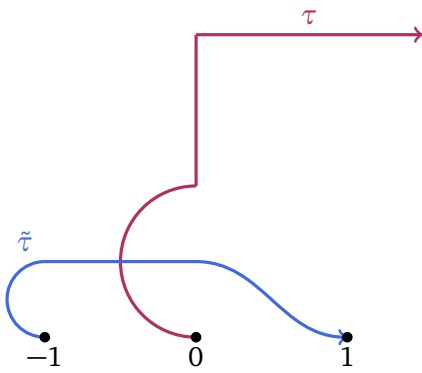

Figure 4: Contours in the holomorphically factorized contour.

The condition $z\tilde{z} < 1$ can be relaxed since the contribution from $z\tilde{z} > 1$ is identical. Indeed, notice that the contour is invariant under $(z, \tilde{z}) \to (\tilde{z}^{-1}, z^{-1})$ and so is $\alpha(z, \tilde{z})$ by (8). Thus we have

$$I = \frac{1}{6}\left(-\int_{z\in[i\delta,1+i\delta]}\int_{\tilde{z}\in[1-i\delta,\infty-i\delta)} + \int_{z\in[-i\delta,1-i\delta]}\int_{\tilde{z}\in[1+i\delta,\infty+i\delta)}\right)\alpha(z, \tilde{z}). \tag{22}$$

We now translate this back to the $\tau$ coordinates. Notice that the $z$-contour running from 0 to 1 maps to a contour connecting 0 and $i\infty$, while the contour in $\tilde{\tau}$ can be pieced together to connect $\tilde{\tau} = -1$ with $\tilde{\tau} = 1$. Thus we simply get

$$I = \frac{1}{6}\int_{\tau\in[0,i\infty)}\int_{\tilde{\tau}\in[-1,1]}\alpha(\tau, \tilde{\tau}). \tag{23}$$

Finally, we modify the contour again appropriately near the cusps to implement the slashed integral (13) or the $i\varepsilon$ prescription. Whenever we approach one of the cusps, we let $\tau$ and $\tilde{\tau}$ turn left into the complex plane. The result is displayed in Figure 4. We exchanged $z$ and $\tilde{z}$ at one point in the derivation which reverses the orientation and thus the sign of the $i\varepsilon$ prescription. Thus the end result in this form is actually real and gives on the nose the slashed integral without having to explicitly take the real part.

## 2.3 Step 3: Decomposing the contour

We will now massage the contour (23), which will allow us to perform the Rademacher expansion in 2.4. To start with these manipulations, let us decompose the contour into two pieces. Let $\Gamma$ be the $\tau$-contour as depicted in Figure 4. We can write

$$6I = \left(\int_{\tau\in\Gamma}\int_{\tilde{\tau}\in-1+\Gamma} - \int_{\tau\in\Gamma}\int_{\tilde{\tau}\in1+\Gamma}\right)\alpha(\tau, \tilde{\tau}). \tag{24}$$

We now rewrite the first term by appropriately deforming the contour. Thus we will simply draw pictures for the contour. Since the integrand is symmetric in $\tau$ and $\tilde{\tau}$, it doesn't matter which contour is for $\tau$ and which for $\tilde{\tau}$, but we still color-code the contours to keep track of the individual steps.

It is also convenient to introduce the notation $a \overset{\text{Re}}{=} b$, which means $\text{Re}\, a = \text{Re}\, b$, since we will in the following often have equalities which are only true for the real part.

### 2.3.1 First term

The first term in (24) can be manipulated as follows,

$$
= \qquad\qquad\qquad\qquad \text{(S-transform)} \tag{25a}
$$

$$
= \qquad\qquad - \qquad\qquad \text{(deform } \tilde{\tau}). \tag{25b}
$$

In the first step we applied the S-transform $\tau \to -1/\tau$ and $\tilde{\tau} \to -1/\tilde{\tau}$ to the joint contour and used invariance of the integrand (6). The second term in (25b) is identical to the second contour in (24) and we learn that

$$
6I = \qquad\qquad - 2 \qquad\qquad . \tag{26}
$$

We will further manipulate the first term in (26) and treat the second contribution below. We first decompose the contour into the part that runs from 0 to $i$ and from $i$ to $i + \infty$,

$$
= \left( \qquad \right) \times \left( \qquad \right) \tag{27a}
$$

$$
= \qquad + \qquad + \qquad + \qquad \tag{27b}
$$

$$
= 2 \qquad + 2 \qquad . \tag{27c}
$$

We used that when splitting both the $\tau$ and $\tilde{\tau}$ contour into two pieces each, two of the four contributions are related by the modular S-transformation. Let us note that the integral

$$
\qquad\qquad , \tag{28}
$$

is purely imaginary. Indeed, we have

$$
\text{(S-transform)} \tag{29a}
$$

$$
\stackrel{\text{Re}}{=} - \quad \text{(horizontal reflection)} \tag{29b}
$$

$$
= - \quad \text{(swap of } \tau \text{ and } \tilde{\tau} \text{)}. \tag{29c}
$$

In the second step, we used the fact that horizontal reflection of the contour gives a minus sign for the real part thanks to the reality property (8). Thus the real part is equal to minus itself and vanishes. We can thus add this contribution to (27c) and learn that

$$
\stackrel{\text{Re}}{=} 2 \qquad + 2 \qquad . \tag{30}
$$

We can further rewrite the second term as follows. Let us expand in the integrand in terms of Fourier modes. The invariance of the integrand under T-modular transformations implies that we can write

$$
\alpha(\tau, \tilde{\tau}) = \sum_{n \in \mathbb{Z}} f_n(\tau + \tilde{\tau}) \exp(\pi i n(\tau - \tilde{\tau})) \, d\tau \wedge d\tilde{\tau}, \tag{31}
$$

for some functions $f_n(z)$. The two reality conditions (8) read in this language

$$
f_n(z) = f_{-n}(z), \qquad f_n(z)^* = -f_n(-\bar{z}). \tag{32}
$$

We then have by direct computation

$$
\begin{aligned}
&= \frac{1}{2} \sum_{n \in \mathbb{Z}} \int_0^\infty ds \, f_n(2i + s) \int_{-s}^s dt \, e^{\pi i n t} \\
&= \sum_{n \in \mathbb{Z}} \int_0^\infty ds \, \frac{\sin(n\pi s)}{n\pi} f_n(2i + s) \\
&\stackrel{\text{Re}}{=} \sum_{n \in \mathbb{Z}} \int_0^\infty ds \, \frac{\sin(n\pi s)}{2n\pi} \left( f_n(2i + s) + f_n(2i + s)^* \right) \\
&= \sum_{n \in \mathbb{Z}} \int_0^\infty ds \, \frac{\sin(n\pi s)}{2n\pi} \left( f_n(2i + s) - f_n(2i - s) \right) \\
&= \sum_{n \in \mathbb{Z}} \int_{-\infty}^\infty ds \, \frac{\sin(n\pi s)}{2n\pi} f_n(2i + s) \\
&= \int_{-\infty}^\infty ds \left[ \frac{s}{2} f_0(2i + s) + \sum_{n \neq 0} \frac{e^{\pi i n s}}{2\pi i n} f_n(2i + s) \right] \\
&\stackrel{\text{Re}}{=} \int_{-\infty}^\infty ds \left[ \frac{s-1}{2} f_0(2i + s) + \sum_{n \neq 0} \frac{e^{\pi i n s}}{2\pi i n} f_n(2i + s) \right]
\end{aligned} \tag{33}
$$

$$= \frac{1}{2} \sum_{n \in \mathbb{Z}} \int_{-\infty}^{\infty} ds \, f_n(2i+s) \int_0^1 dx \, (s-2x) \, e^{\pi i n(s-2x)}$$

$$= \frac{1}{2} \int_{i-\infty}^{i+\infty} d\tau \int_i^{i+1} d\tilde{\tau} \, (\tau - \tilde{\tau}) \sum_{n \in \mathbb{Z}} f_n(\tau + \tilde{\tau}) \exp(\pi i n(\tau - \tilde{\tau}))$$

$$= \frac{1}{2}(\tau - \tilde{\tau}) \qquad .$$

Here $(\tau - \tilde{\tau})$ means that we are multiplying the integrand by $(\tau - \tilde{\tau})$ before integrating. Since the integrand is no longer symmetric in $\tau$ and $\tilde{\tau}$, we added the labels. Thus we can write

$$\overset{\text{Re}}{=} 2 \qquad + (\tau - \tilde{\tau}) \qquad \qquad (34a)$$

$$= (\tau - \tilde{\tau}) \qquad + (\tau - \tilde{\tau}) \qquad \qquad (34b)$$

$$= (\tau - \tilde{\tau}) \qquad \qquad . \qquad \qquad (34c)$$

Here we used that the difference of the contour over the two semicircles in the first term partially cancels thanks to invariance of the integrand under T-modular transformations.

### 2.3.2 Second term

We can further massage the second contribution in (26) as follows

$$= \qquad + \qquad \text{(deform } \tau\text{)} \qquad (35a)$$

$$= \qquad + \qquad + \qquad \text{(deform } \tilde{\tau}\text{)} \qquad (35b)$$

$$= \qquad + \qquad + \qquad \text{(T-transform 3}^{\text{rd}} \text{ term)} \quad (35c)$$

$$= \qquad + \qquad + \qquad \text{(S-transform 2}^{\text{nd}} \text{ and 3}^{\text{rd}} \text{ term)} \qquad (35d)$$

$$\overset{\text{Re}}{=} \;\; \text{[contour diagram]} \;\; + \;\; \text{[contour diagram]} \qquad (2^{\text{nd}} \text{ term imaginary}) \tag{35e}$$

$$\overset{\text{Re}}{=} \;\; \text{[contour diagram]} \;\; + \;\; \text{[contour diagram]} \;\; + \;\; \text{[contour diagram]} \qquad (\text{deform } \tau) \tag{35f}$$

$$\overset{\text{Re}}{=} \;\; \text{[contour diagram]} \qquad (2^{\text{nd}} \text{ and } 3^{\text{rd}} \text{ terms imaginary}) . \tag{35g}$$

We first decomposed the integral. From (35b) to (35c), we used invariance under the T-modular transformation ($\tau \to \tau + 1$ and $\tilde{\tau} \to \tilde{\tau} - 1$) of the $3^{\text{rd}}$ term. We then applied the S-modular transformation ($\tau \to -1/\tau$ and $\tilde{\tau} \to -1/\tilde{\tau}$) to the last two terms to obtain (35d). We then used that the second contribution integral in (35d) is purely imaginary. This follows from the fact that we can reflect the contour horizontally at the cost of a minus sign (for the real part), which in turn is a consequence of the second reality condition in (8). By applying the following series of contour deformations, this implies that this contribution is purely imaginary,

$$\text{[contour diagram]} \;\; = \;\; \text{[contour diagram]} \qquad (\text{deform } \tilde{\tau}) \tag{36a}$$

$$= \;\; \text{[contour diagram]} \qquad (\text{deform } \tilde{\tau}) \tag{36b}$$

$$\overset{\text{Re}}{=} - \;\; \text{[contour diagram]} \qquad (\text{horizontal reflection}) \tag{36c}$$

$$= - \;\; \text{[contour diagram]} \qquad (\text{reverse orientation}) \tag{36d}$$

$$\overset{\text{Re}}{=} 0 . \tag{36e}$$

The contributions of the two arcs in the first line (36a) cancel thanks to T-modular invariance. We use the same logic to see that the second term in (35e) is purely imaginary. The last term in (35f) is also purely imaginary since we can reflect it along the vertical axis at the cost of a minus sign, but the contour is invariant.

### 2.3.3 Summary

Using the manipulated expressions (34c) and (35g) for the first and second contribution, respectively, (26) becomes

$$I \overset{\text{Re}}{=} \frac{1}{6}(\tau - \tilde{\tau}) \;\; \text{[contour diagram]} \;\; - \frac{1}{3} \;\; \text{[contour diagram]} . \tag{37}$$

We will apply the Rademacher procedure to this contour. These two contributions behave quite differently.

## 2.4 Step 4: Rademacherization

We will now deform the $\tau$ and $\tilde{\tau}$ contours in (37) into Ford circles, which will lead to a two-dimensional version of the Rademacher contour. This deformation can be achieved in a series of steps. We start by reviewing the standard Rademacher contour.

### 2.4.1 Farey sequence, Ford circles and all that

We start by defining the *Farey sequence* $F_n$ consisting of all irreducible fractions $0 \leq \frac{a}{c} \leq 1$ where the denominator is bounded, $c \leq n$. A fraction is irreducible if $(a, c) = 1$, i.e., $a$ and $c$ are coprime. While conventions differ in the literature, we do include both $\frac{0}{1}$ and $\frac{1}{1}$ in the sequence. The first few sequences are

$$F_1 = \left( \tfrac{0}{1}, \tfrac{1}{1} \right), \tag{38a}$$

$$F_2 = \left( \tfrac{0}{1}, \tfrac{1}{2}, \tfrac{1}{1} \right), \tag{38b}$$

$$F_3 = \left( \tfrac{0}{1}, \tfrac{1}{3}, \tfrac{1}{2}, \tfrac{2}{3}, \tfrac{1}{1} \right), \tag{38c}$$

$$F_4 = \left( \tfrac{0}{1}, \tfrac{1}{4}, \tfrac{1}{3}, \tfrac{1}{2}, \tfrac{2}{3}, \tfrac{3}{4}, \tfrac{1}{1} \right), \tag{38d}$$

$$F_5 = \left( \tfrac{0}{1}, \tfrac{1}{5}, \tfrac{1}{4}, \tfrac{1}{3}, \tfrac{2}{5}, \tfrac{1}{2}, \tfrac{3}{5}, \tfrac{2}{3}, \tfrac{3}{4}, \tfrac{4}{5}, \tfrac{1}{1} \right). \tag{38e}$$

In the limit as $n \to \infty$, we generate all irreducible fractions in the range $[0, 1]$ that densely cover this interval.

For each $\frac{a}{c}$, we are going to introduce a *Ford circle* $C_{a/c}$ in the $\tilde{\tau}$-plane, which is anchored at the point $\tilde{\tau} = \frac{a}{c}$ on the real axis and has radius $\frac{1}{2c^2}$ (meaning that the center is at $\tilde{\tau} = \frac{a}{c} + \frac{i}{2c^2}$). We give each $C_{a/c}$ a clockwise orientation. One can show that two circles $C_{a/c}$ and $C_{a'/c'}$ intersect at a point only if $\frac{a}{c}$ and $\frac{a'}{c'}$ are two neighboring fractions in any Farey sequence $F_n$. Hence for any $n$, $F_n$ gives a sequence of circles where every neighboring pair touches at a unique point.

In the particular, the $\tilde{\tau}$ contour in (37) can be understood as starting at $\tilde{\tau} = 0$, following the Ford circle $C_{0/1}$ clockwise until it reaches $C_{1/1}$, and then continuing along $C_{1/1}$ counter-clockwise until the point $\tilde{\tau} = 1$. Let us call this contour $\Gamma_1$, since it followed the Ford circles in the Farey sequence $F_1$. We can now introduce a sequence of *Rademacher contours* $\Gamma_n$ which are all deformations of $\Gamma_1$. Each $\Gamma_n$ follows the sequence of Ford circles prescribed by $F_n$ in a generalization of the above procedure [38, 39].

In the end, we take $n \to \infty$ and obtain a sum over all Ford circles $C_{a/c}$ with $0 \leq \frac{a}{c} < 1$ (the circle $C_{1/1}$ drops out in the limit). The corresponding Rademacher contour $\Gamma_\infty$ is illustrated in Figure 5.

### 2.4.2 Rademacherization of the first contribution

We can immediately apply this contour deformation to the $\tilde{\tau}$ contour of the first contribution in (37), leading to

$$\frac{1}{6}(\tau - \tilde{\tau}) \quad \underset{0 \qquad 1}{\overset{\tilde{\tau}}{\curvearrowright}} \quad {}^{\tau} = \frac{1}{6} \sum_{c=1}^{\infty} \sum_{\substack{a=0 \\ (a,c)=1}}^{c-1} \int_{\tau \in \longrightarrow} \int_{\tilde{\tau} \in C_{a/c}} (\tau - \tilde{\tau}) \alpha(\tau, \tilde{\tau}). \tag{39}$$

This contour is illustrated in Figure 5. Convergence of this Rademacher expansion will be analyzed more carefully in appendix B.

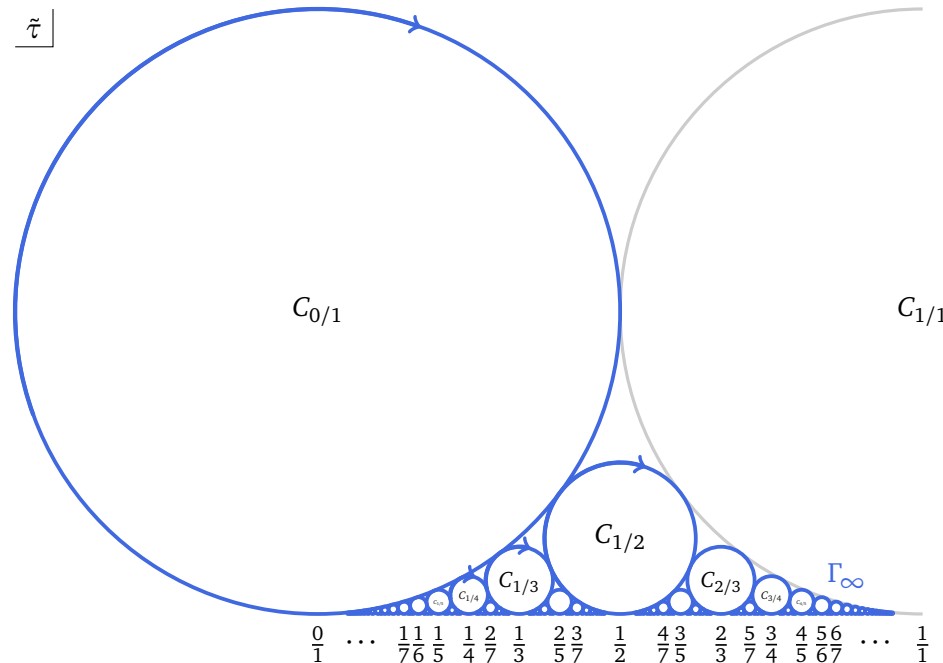

Figure 5: Rademacher contour $\Gamma_\infty$ in the $\tilde{\tau}$-plane, enclosing Ford circles $C_{a/c}$ for all irreducible fractions $\frac{a}{c} \in [0, 1)$.

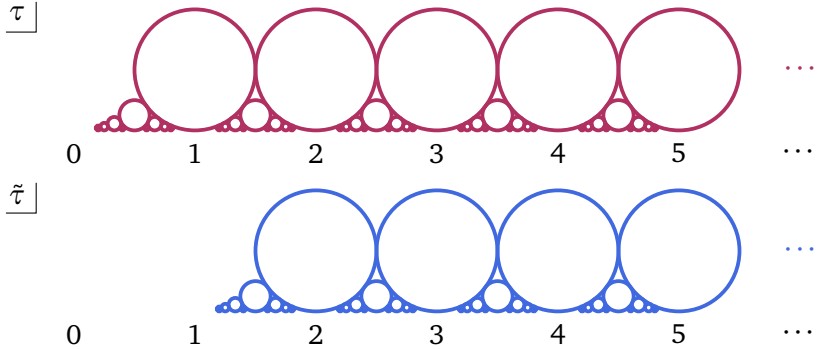

Figure 6: Ford circles in the $\tau$ and $\tilde{\tau}$ upper half-planes used in the double Rademacher contour (40b).

### 2.4.3 Rademacherization of the second contribution

We next deform the second contribution in (37) into a 'double Rademacher contour', meaning that both the contour over $\tau$ and $\tilde{\tau}$ will be deformed into a sum over Ford circles $C_{a/c}$ and $C_{\tilde{a}/\tilde{c}}$ with $\frac{a}{c} > 0$ and $\frac{\tilde{a}}{\tilde{c}} > 1$, respectively. Leaving the issue of convergence momentarily aside, this leads to

$$-\frac{1}{3} \int = -\frac{1}{3} \sum_{c=1}^\infty \sum_{\tilde{c}=1}^\infty \sum_{\substack{a=1 \\ (a,c)=1}}^\infty \sum_{\substack{\tilde{a}=\tilde{c}+1 \\ (\tilde{a},\tilde{c})=1}}^\infty \int_{C_{a/c}} \mathrm{d}\tau \int_{C_{\tilde{a}/\tilde{c}}} \mathrm{d}\tilde{\tau}\ \alpha(\tau, \tilde{\tau}) \tag{40a}$$

$$= -\frac{1}{3} \sum_{c=1}^\infty \sum_{\tilde{c}=1}^\infty \sum_{\substack{a=1 \\ (a,c)=1}}^\infty \sum_{\substack{\tilde{a}=\tilde{c}+1 \\ (\tilde{a},\tilde{c})=1}}^\infty \int_{C_{a/\gamma}} \mathrm{d}\tau \int_{C_\infty} \mathrm{d}\tilde{\tau}\ \alpha(\tau, \tilde{\tau}). \tag{40b}$$

The contours used in the above double sum are illustrated in Figure 6. In the second line, we map $\frac{\tilde{a}}{\tilde{c}}$ to infinity by applying a joint modular transformation in $\tau$ and $\tilde{\tau}$. This maps the Ford circle at $\frac{a}{c}$ into $\frac{\alpha}{\gamma}$, where

$$\frac{\alpha}{\gamma} = \frac{a\tilde{d} + \tilde{b}c}{\tilde{a}c + a\tilde{c}} \,. \tag{41}$$

Here, $\tilde{b}$ and $\tilde{d}$ are determined such that $\tilde{a}\tilde{d} - \tilde{b}\tilde{c} = 1$. This determines $\frac{\alpha}{\gamma}$ up to the addition of an integer which doesn't influence the above integral. We can thus reorganize the sum into a sum over $\alpha$ and $\gamma$ and count how many times this fraction appears in the above sum. Let us denote this number by $m(\frac{\alpha}{\gamma})$, which we will determine below. It will in particular turn out to be finite. Thus we can write

$$-\frac{1}{3} \int\!\!\int = -\frac{1}{3} \sum_{\gamma=1}^{\infty} \sum_{\substack{\alpha=0 \\ (\alpha,\gamma)=1}}^{\gamma-1} m\left(\frac{\alpha}{\gamma}\right) \int_{C_\infty} \mathrm{d}\tau \int_{C_{\alpha/\gamma}} \mathrm{d}\tilde{\tau} \, \alpha(\tau,\tilde{\tau}) \tag{42a}$$

$$= \frac{1}{3} \sum_{c=1}^{\infty} \sum_{\substack{a=0 \\ (a,c)=1}}^{c-1} m\left(\frac{a}{c}\right) \, \bigcirc \, , \tag{42b}$$

where we changed the notation back to $a$ and $c$. The extra minus sign originates from the fact that the contour $C_\infty$ is a horizontal contour running to the left.

To fully evaluate the Rademacher expansion it remains to find a useful formula for $m(\frac{\alpha}{\gamma})$. Its definition is

$$m\left(\frac{\alpha}{\gamma}\right) = \#\left\{ \frac{a}{c} \in \mathbb{Q}_{>0} \,,\; \frac{\tilde{a}}{\tilde{c}} \in \mathbb{Q}_{>1} \; \middle| \; \frac{\alpha}{\gamma} \equiv \frac{a\tilde{d} + \tilde{b}c}{\tilde{a}c + a\tilde{c}} \bmod 1 \right\} , \tag{43}$$

where $\tilde{b}$ and $\tilde{d}$ are determined such that $\tilde{a}\tilde{d} - \tilde{b}\tilde{c} = 1$. It is easy to implement this numerically and find the first few values of $m(\frac{\alpha}{\gamma})$, see Table 1.

From the table, one can conjecture that the following recursion relation holds:

$$m\left(\frac{a}{c}\right) = \begin{cases} m\left(\frac{a}{c-a}\right), & 2a < c \,, \\ m\left(\frac{2a-c}{a}\right) + 1, & 2a > c \,. \end{cases} \tag{44}$$

This determines $m(\frac{a}{c})$ completely together with the initial conditions $m(\frac{1}{2}) = m(0) = 0$. We prove in appendix A that this recursion formula holds indeed.

### 2.4.4 Summary

Overall, we hence derived

$$I \overset{\mathrm{Re}}{=} \frac{1}{6} \sum_{c=1}^{\infty} \sum_{\substack{a=0 \\ (a,c)=1}}^{c-1} \int_{\tau \in \longrightarrow} \int_{\tilde{\tau} \in C_{a/c}} \left( \tau - \tilde{\tau} + 2m\left(\frac{a}{c}\right) \right) \alpha(\tau, \tilde{\tau}) \,. \tag{45}$$

The contour deformations are done at this point. We will massage the formula a bit further and take the real part explicitly by adding the complex conjugate and using the reality property (8).

Table 1: The first few values of $m\left(\frac{a}{c}\right)$.

| a \ c | 0 | 1 | 2 | 3 | 4 | 5 | 6 | 7 | 8 | 9 | 10 | 11 | 12 | 13 | 14 | 15 |
|---|---|---|---|---|---|---|---|---|---|---|---|---|---|---|---|---|
| 1 | 0 | | | | | | | | | | | | | | | |
| 2 | | 0 | | | | | | | | | | | | | | |
| 3 | | 0 | 1 | | | | | | | | | | | | | |
| 4 | | 0 | | 2 | | | | | | | | | | | | |
| 5 | | 0 | 1 | 1 | 3 | | | | | | | | | | | |
| 6 | | 0 | | | | 4 | | | | | | | | | | |
| 7 | | 0 | 1 | 2 | 1 | 2 | 5 | | | | | | | | | |
| 8 | | 0 | | 1 | | 2 | | 6 | | | | | | | | |
| 9 | | 0 | 1 | | 3 | 1 | | 3 | 7 | | | | | | | |
| 10 | | 0 | | 2 | | | | 2 | | 8 | | | | | | |
| 11 | | 0 | 1 | 1 | 1 | 4 | 1 | 3 | 3 | 4 | 9 | | | | | |
| 12 | | 0 | | | | 2 | | 2 | | | | 10 | | | | |
| 13 | | 0 | 1 | 2 | 3 | 2 | 5 | 1 | 2 | 2 | 3 | 5 | 11 | | | |
| 14 | | 0 | | 1 | | 1 | | | | 4 | | 4 | | 12 | | |
| 15 | | 0 | 1 | | 1 | | | 6 | 1 | | | 4 | | 6 | 13 | |
| 16 | | 0 | | 2 | | 4 | | 3 | | 2 | | 2 | | 4 | | 14 |

This gives

$$
I = \frac{1}{12} \sum_{c=1}^{\infty} \sum_{\substack{a=0 \\ (a,c)=1}}^{c-1} \left[ \int_{\tau \in \longrightarrow} \int_{\tilde{\tau} \in C_{a/c}} \left[ \tau - \tilde{\tau} + 2m\left(\frac{a}{c}\right) \right] \alpha(\tau, \tilde{\tau}) \right.
$$
$$
\left. - \int_{-\tilde{\tau} \in \longrightarrow} \int_{-\tilde{\tilde{\tau}} \in C_{-a/c}} \left[ \bar{\tau} - \tilde{\bar{\tau}} + 2m\left(\frac{a}{c}\right) \right] \alpha(-\bar{\tau}, -\tilde{\bar{\tau}}) \right] \tag{46a}
$$

$$
= \frac{1}{12} \sum_{c=1}^{\infty} \sum_{\substack{a=0 \\ (a,c)=1}}^{c-1} \left[ \int_{\tau \in \longrightarrow} \int_{\tilde{\tau} \in C_{a/c}} \left[ \tau - \tilde{\tau} + 2m\left(\frac{a}{c}\right) \right] \alpha(\tau, \tilde{\tau}) \right.
$$
$$
\left. + \int_{\tau \in \longrightarrow} \int_{\tilde{\tau} \in C_{-a/c}} \left[ \tau - \tilde{\tau} - 2m\left(\frac{a}{c}\right) \right] \alpha(\tau, \tilde{\tau}) \right] \tag{46b}
$$

$$
= \frac{1}{6} \int_{\tau \in \longrightarrow} \int_{\tilde{\tau} \in C_0} (\tau - \tilde{\tau}) \alpha(\tau, \tilde{\tau})
$$
$$
+ \frac{1}{12} \sum_{c=1}^{\infty} \sum_{\substack{a=1 \\ (a,c)=1}}^{c-1} \left[ \int_{\tau \in \longrightarrow} \int_{\tilde{\tau} \in C_{a/c}} \left[ \tau - \tilde{\tau} + 2m\left(\frac{a}{c}\right) \right] \alpha(\tau, \tilde{\tau}) \right.
$$
$$
\left. + \int_{\tau \in \longrightarrow} \int_{\tilde{\tau} \in C_{1-a/c}} \left[ \tau - \tilde{\tau} + 2 - 2m\left(\frac{a}{c}\right) \right] \alpha(\tau + 1, \tilde{\tau} - 1) \right] \tag{46c}
$$

$$
= \frac{1}{6} \sum_{c=1}^{\infty} \sum_{\substack{a=0 \\ (a,c)=1}}^{c-1} \int_{\tau \in \longrightarrow} \int_{\tilde{\tau} \in C_{a/c}} \left[ \tau - \tilde{\tau} + \mu\left(\frac{a}{c}\right) \right] \alpha(\tau, \tilde{\tau}). \tag{46d}
$$

In the last line, we renamed $a \to c - a$ for the last term and defined for $x \in \mathbb{Q} \cap [0, 1)$

$$\mu(x) = \begin{cases} 0, & x = 0, \\ m(x) - m(1-x) + 1, & 0 < x < 1. \end{cases} \tag{47}$$

As a consequence of the recursion relation for $m$ in (44), $\mu$ satisfies the recursion relation

$$\mu\left(\frac{a}{c}\right) = \begin{cases} \mu\left(\frac{a}{c-a}\right) - 1, & 2a < c, \\ \mu\left(\frac{2a-c}{a}\right) + 1, & 2a \geqslant c, \end{cases} \tag{48}$$

with initial condition $\mu(0) = 0$.

## 2.5 Step 5: Relating to Dedekind sums

We now relate $\mu$ to more standard number theoretic objects.

### 2.5.1 Dedekind sums

For coprime integers $(c, d)$ with $c > 0$, the Dedekind sum is defined as

$$s(d, c) = \sum_{k=1}^{c-1} \left(\!\!\left(\frac{k}{c}\right)\!\!\right) \left(\!\!\left(\frac{dk}{c}\right)\!\!\right), \tag{49}$$

where $((x)) := x - \lfloor x \rfloor - \frac{1}{2}$ for $x \notin \mathbb{Z}$, and $((x)) := 0$ for $x \in \mathbb{Z}$. The Rademacher function $\Phi : \mathrm{SL}(2, \mathbb{Z}) \to \mathbb{Z}$ is

$$\Phi \begin{pmatrix} a & b \\ c & d \end{pmatrix} = \begin{cases} \frac{b}{d}, & \text{for } c = 0, \\ \frac{a+d}{c} - 12(\text{sign } c)s(d, |c|), & \text{for } c \neq 0. \end{cases} \tag{50}$$

Dedekind sums and the Rademacher function appear in various contexts; see [40] for a review. In particular the Rademacher function is the phase in the modular transformation of the $\eta$-function,

$$\log \eta(\gamma \cdot \tau) = \log \eta(\tau) + \frac{1}{2}(\text{sign } c)^2 \log\left(\frac{c\tau + d}{i \,\text{sign } c}\right) + \frac{i\pi}{12}\Phi(\gamma), \tag{51}$$

with $\gamma = \begin{pmatrix} a & b \\ c & d \end{pmatrix}$ and $\text{sign } 0 := 0$.

The Rademacher function satisfies the composition rule [40, p.51]

$$\Phi(\gamma_2 \gamma_1) = \Phi(\gamma_2) + \Phi(\gamma_1) - 3\,\text{sign}(c_1 c_2 c_3), \tag{52}$$

where $c_3 = c(\gamma_2 \gamma_1)$. This rule and the initial conditions $\Phi(S) = \Phi(\text{id}) = 0$, $\Phi(T) = 1$ characterize it uniquely. For $\gamma = \begin{pmatrix} a & b \\ c & d \end{pmatrix}$ with $0 < a < c$, two special cases relevant for the upcoming section are

$$\Phi(\gamma) = \Phi(A\gamma) - 1, \tag{53a}$$
$$\Phi(\gamma) = \Phi(B\gamma) + 1, \tag{53b}$$

where $A = (TST)^{-1} = \begin{pmatrix} 1 & 0 \\ -1 & 1 \end{pmatrix}$ and $B = T^2 S = \begin{pmatrix} 2 & -1 \\ 1 & 0 \end{pmatrix}$.

### 2.5.2 Relating $\mu$ to the Rademacher function

We claim that

$$\mu\left(\frac{a}{c}\right) = \Phi\begin{pmatrix} a & b \\ c & d \end{pmatrix}, \tag{54}$$

for $0 \leqslant a < c$. On the right-hand side, $d = a^* \in \{0, 1, \dots, c-1\}$ is the modular inverse of $a \bmod c$ and $b$ is uniquely determined through $ad - bc = 1$.

To demonstrate (54), we temporarily denote the RHS by $\hat{\mu}(\frac{a}{c})$ and will show that $\mu(\frac{a}{c}) = \hat{\mu}(\frac{a}{c})$. It satisfies

$$\hat{\mu}(0) = \Phi\begin{pmatrix} 0 & -1 \\ 1 & 0 \end{pmatrix} = \Phi(S) = 0. \tag{55}$$

The two equations (53) imply recursion relations for $\hat{\mu}$. For $0 < 2a < c$, (53a) becomes

$$\hat{\mu}\left(\frac{a}{c}\right) = \Phi\begin{pmatrix} a & b \\ c-a & d-b \end{pmatrix} - 1 = \hat{\mu}\left(\frac{a}{c-a}\right) - 1, \quad \text{if} \quad 0 \leqslant d - b < c - a. \tag{56}$$

The restriction on $a$ is necessary so that $0 < a < (c-a)$ and the argument of $\hat{\mu}$ on the right hand side falls in the appropriate range. The additional condition in (56) ensures that lower right entry of $\Phi$ falls in the range as explained below (54). However, it is automatically satisfied and can be removed:

$$d - b = \frac{1 + (c-a)d}{c} > \frac{(c-a)d}{c} > 0, \tag{57}$$

$$c - a - (d-b) = \frac{(c-a)(c-d) - 1}{c} > \frac{2 \cdot 1 - 1}{c} > 0. \tag{58}$$

For $c \leqslant 2a < 2c$, (53b) becomes

$$\hat{\mu}\left(\frac{a}{c}\right) = \Phi\begin{pmatrix} 2a-c & 2b-d \\ a & b \end{pmatrix} + 1 = \hat{\mu}\left(\frac{2a-c}{a}\right) + 1, \quad \text{if} \quad 0 \leqslant b < a. \tag{59}$$

The additional inequality is again superfluous since we have automatically

$$b = \frac{ad - 1}{c} \geqslant 0, \tag{60}$$

$$a - b = \frac{1 + a(c-d)}{c} > 0. \tag{61}$$

The two equations (56) and (61) show that $\hat{\mu}(\frac{a}{c})$ satisfies the recursion relation (48). Since also the initial condition (55) coincides and the recursion determines $\hat{\mu}(\frac{a}{c})$ uniquely, we conclude that $\mu(\frac{a}{c}) = \hat{\mu}(\frac{a}{c})$ as claimed.

### 2.5.3 Finishing the derivation

We are now almost done. Let us symmetrize the contributions in (46d) from $C_{a/c}$ and $C_{a^*/c}$. Notice that with $d = a^*$

$$\int_{\tau \in C_\infty} \int_{\tilde{\tau} \in C_{d/c}} \alpha(\tau, \tilde{\tau}) = \int_{\tau \in C_\infty} \int_{\tilde{\tau} \in C_{d/c}} \alpha\left(\frac{a\tau + b}{c\tau + d}, \frac{a\tilde{\tau} - b}{-c\tilde{\tau} + d}\right) \tag{62a}$$

$$= \int_{C_{a/c}} \int_{C_\infty} \alpha(\tau, \tilde{\tau}) \tag{62b}$$

$$= \int_{C_\infty} \int_{C_{a/c}} \alpha(\tau, \tilde{\tau}), \tag{62c}$$

where we used modular invariance and the symmetry under $\tau \leftrightarrow \tilde{\tau}$ exchange (8). Thus we can write

$$I = \frac{1}{6} \sum_{c=1}^{\infty} \sum_{\substack{a=0 \\ (a,c)=1}}^{c-1} \int_{\tau \in \longrightarrow} \int_{\tilde{\tau} \in C_{a/c}} \left( \tau - \tilde{\tau} + \frac{2a}{c} + \mu\left(\frac{a}{c}\right) - \frac{a+a^*}{c} \right) \alpha(\tau, \tilde{\tau}) \tag{63a}$$

$$= \sum_{c=1}^{\infty} \sum_{\substack{a=0 \\ (a,c)=1}}^{c-1} \int_{\tau \in \longrightarrow} \int_{\tilde{\tau} \in C_{a/c}} \left[ \frac{1}{6}\left( \tau - \tilde{\tau} + \frac{2a}{c} \right) - 2s(a,c) \right] \alpha(\tau, \tilde{\tau}). \tag{63b}$$

We used that as a consequence of the discussion above

$$\mu\left(\frac{a}{c}\right) = \Phi\begin{pmatrix} a & b \\ c & a^* \end{pmatrix} = \frac{a+a^*}{c} - 12s(a^*, c) = \frac{a+a^*}{c} - 12s(a, c). \tag{64}$$

## 2.6 Further comments

### 2.6.1 Why step 3 was necessary

It may not be immediately obvious why step 3 in section 2.3 was necessary to apply the Rademacher procedure. One may try to directly apply the Rademacher expansion to the contour in Figure 4. This leads to two problems. Similar to the discussion in section 2.4, we would want to apply the Rademacher expansion to both the $\tau$ and $\tilde{\tau}$ contour and then reorganize the sum into a single sum over cusps with multiplicities. Contrary to the multiplicities $m(\frac{a}{c})$ that we encountered in step 4, these multiplicities turn out to be infinite and one gets nonsensical answers. A related problem is the occurrence of the cusp $(\tau, \tilde{\tau}) \in C_{a/c} \times C_{-a/c}$ in the expansion. This cusp can be mapped via a joint modular transformation to $C_{\infty} \times C_{\infty}$, i.e. the product of two horizontal contours. The integral over this contour also diverges because of T-modular invariance $\alpha(\tau - 1, \tilde{\tau} + 1) = \alpha(\tau, \tilde{\tau})$,

$$\int_{\tau \in C_{a/c}} \int_{\tilde{\tau} \in C_{-a/c}} \alpha(\tau, \tilde{\tau}) = \int_{\tau \in \longrightarrow} \int_{\tilde{\tau} \in \longrightarrow} \alpha(\tau, \tilde{\tau}) = \infty. \tag{65}$$

The contour deformations in step 3 were designed so that these problems do not appear when we apply the Rademacher expansion.

### 2.6.2 Other Lorentzian integration formulae

As we already mentioned in the introduction, there are several small variations on the Lorentzian integration formula (23). The form that we presented originates from picking the $\Gamma(2)$ subgroup of the modular group. One can map the fundamental domain to the sphere in other ways. For example, every other genus 0 subgroup of the modular group gives a variation of the formula. One can also map the fundamental domain to parts of the complex plane and apply similar contour deformation logic. We show in appendix C that one can also derive

$$\oint_{\mathcal{F}} \alpha(\tau, \tilde{\tau}) = \frac{1}{6} \int_{\tau \in [0, i\infty)} \int_{\tilde{\tau} \in [-2,2]} \alpha(\tau, \tilde{\tau}) \tag{66a}$$

$$= \frac{1}{6} \int_{\tau \in [0, i\infty)} \int_{\tilde{\tau} \in [-\frac{1}{2}, \frac{1}{2}]} \alpha(\tau, \tilde{\tau}) \tag{66b}$$

$$\overset{\text{Re}}{=} \frac{1}{2} \int_{\tau \in [0, i\infty)} \int_{\tilde{\tau} \in [e^{\frac{2\pi i}{3}}, e^{\frac{\pi i}{3}}]} \alpha(\tau, \tilde{\tau}). \tag{66c}$$

The equations (66a) and (66b) are also derived from $\Gamma(2)$, but using different coordinates, while (66c) is directly derived using the Hauptmodul of the full modular group. Near the cusps, the contour is appropriately modified to implement the $i\varepsilon$ as discussed in section 2.2. We expect that there are many other variations of such formulas.

Clearly (66c) is less adapted as a starting point for the Rademacher procedure since the contour does not end at the cusps. One can however derive a Rademacher formula from the other formulas and we do this in appendix C for (66a). After the dust has settled, we find exactly the same formula as (2) and thus the Rademacher formula seems to be canonical. This is perhaps not surprising since otherwise we could take the difference of the two formulas and learn a relation that has to be satisfied by the polar terms of all integrands.

### 2.6.3  The imaginary part and the $i\varepsilon$ prescription

We explained our prescription for the regularized integral over the fundamental domain in section 2.1, where we averaged the two possibilities for the $i\varepsilon$ prescription. In the string theory context, there is a preferred sign of the $i\varepsilon$ prescription. One can also incorporate it in the Rademacher formula (even though the imaginary part does have a closed form evaluation). Indeed, we can compute the imaginary part as half of the difference between both $i\varepsilon$ prescriptions. This gives

$$\int_{\mathcal{F}_{i\varepsilon}} \alpha(\tau,\tilde\tau) = -\frac{1}{2} \int_{x\in[-\frac{1}{2},\frac{1}{2}]} \int_{y\in iL+\mathbb{R}} \alpha(x,y) \tag{67a}$$

$$= -\frac{1}{2} \int_{\tau\in\longrightarrow} \int_{\tilde\tau\in[i,i+1]} \alpha(\tau,\tilde\tau) \tag{67b}$$

$$= -\frac{1}{2} \sum_{c=1}^{\infty} \sum_{\substack{a=0 \\ (a,c)=1}}^{c-1} \int_{\tau\in\longrightarrow} \int_{\tilde\tau\in C_{a/c}} \alpha(\tau,\tilde\tau). \tag{67c}$$

Thus, one can modify the Rademacher formula (2) to correspond to the string theoretic $i\varepsilon$-prescription as follows

$$\int_{\mathcal{F}_{i\varepsilon}} \alpha(\tau,\tilde\tau) = \sum_{c=1}^{\infty} \sum_{\substack{a=0 \\ (a,c)=1}}^{c-1} \int_{\tau\in\longrightarrow} \int_{\tilde\tau\in C_{a/c}} \left[ \frac{1}{6}\left(\tau - \tilde\tau + \frac{2a}{c}\right) - 2s(a,c) - \frac{1}{2} \right] \alpha(\tau,\tilde\tau). \tag{68}$$

## 3  Applications

To illustrate the Rademacher formula (2), we now work out a number of examples.

### 3.1  Reduction to polar terms

In the applications we are going to illustrate, there are a set of common recurring steps. Therefore, we will devote this subsection to illustrating them, in order to avoid repeating ourselves. We will often deal with $\alpha(\tau,\tilde\tau)$ of the form

$$\alpha(\tau,\tilde\tau) = \frac{\mathcal{N}}{2i} \frac{d\tau \wedge d\tilde\tau}{\operatorname{Im}(\tau)^n} f(\tau)^{\mathsf{T}} g(\tilde\tau), \tag{69}$$

with $\mathcal{N}$ some constant and $g(\tilde\tau)$ some vector-valued modular form that, under modular transformations, transforms as

$$g\left(\frac{a\tilde\tau + b}{c\tilde\tau + d}\right) = \frac{1}{(-i(c\tilde\tau + d))^{n-2}} \rho_{a,d,c}\, g(\tilde\tau), \tag{70}$$

with $\rho_{a,d,c}$ an $m \times m$ matrix acting on the $m$-vector $g$, depending on the SL(2, $\mathbb{Z}$) matrix $\left(\begin{smallmatrix} a & b \\ c & d \end{smallmatrix}\right)$.[6] $f$ satisfies an identical transformation law with $\rho_{a,d,c}$ replaced by $(\rho_{-a,-d,c}^{-1})^{\mathsf{T}}$, so that $\alpha$ is modular invariant as in (6). Notice that (69) satisfies the convergence criterion (B.1) for $n > 2$, i.e. when the weight of $f$ and $g$ is negative. We will also consider the case $n = 2$ in section 3.6, where we can still have conditional convergence.

In order to evaluate the sum (63a), we will have to compute an integral of the form

$$I_{a/c} := \frac{2^{n-1}\mathcal{N}}{6i} \int_{\longrightarrow} \mathrm{d}\tau \int_{C_{a/c}} \frac{\mathrm{d}\tilde{\tau}}{(-i(\tau + \tilde{\tau}))^n} \left(\tau - \tilde{\tau} + \frac{2a}{c} - 12s(a,c)\right) f(\tau)^{\mathsf{T}} g(\tilde{\tau}). \quad (71)$$

At this point, we change variables: $\tilde{\tau} \to \frac{a\tilde{\tau}+b}{c\tilde{\tau}+d}$ in the $\tilde{\tau}$ integral to map $C_{a/c}$ to $\longrightarrow$. This gives an extra minus sign from the orientation reversal of the contour. Including the Jacobian and the modular transformation behaviour of $g$ yields

$$I_{a/c} = \frac{2^{n-1}\mathcal{N}}{6i} \int_{\longrightarrow} \mathrm{d}\tau \int_{\longrightarrow} \frac{\mathrm{d}\tilde{\tau}\, f(\tau)^{\mathsf{T}} \rho_{a,d,c}\, g(\tilde{\tau})}{(-(a\tilde{\tau}+b+\tau(c\tilde{\tau}+d)))^n} \left(\tau - \frac{a\tilde{\tau}+b}{c\tilde{\tau}+d} + \frac{2a}{c} - 12s(a,c)\right). \quad (72)$$

We now use the following fact: in all the cases we will consider, only terms that have a cusp expansion of the form $e^{-ix\tau - i\tilde{x}\tilde{\tau}}$ with $x, \tilde{x} > 0$ will contribute to the integral. Indeed, if $x < 0$ or $\tilde{x} < 0$, we can make the integral arbitrarily small by pushing the horizontal contour in $\tau$ or $\tilde{\tau}$ to large imaginary values. Thus the integral vanishes for such terms. These terms are the generalization of polar terms of holomorphic modular objects. Therefore, we can substitute in our integral

$$f(\tau)^{\mathsf{T}} \rho_{a,d,c} g(\tilde{\tau}) \longrightarrow \sum_{k,\ell} \sum_{\substack{x \in \mathrm{Polar}_k(f) \\ \tilde{x} \in \mathrm{Polar}_\ell(g)}} \rho_{a,d,c}^{k,\ell} M^k(x) \tilde{M}^\ell(\tilde{x}) e^{-ix\tau - i\tilde{x}\tilde{\tau}}, \quad (73)$$

where $k$ and $\ell$ label the columns and rows of the matrix $\rho_{a,d,c}$ and $x$ and $\tilde{x}$ run over all polar terms of the components of $f$ and $g$ with coefficients given by $M^k(x)$ and $\tilde{M}^\ell(\tilde{x})$, respectively. In our examples, the sum in (73) only contains a few terms and we will in the following just consider a single term $\rho_{a,d,c}^{k,\ell} e^{-ix\tau - i\tilde{x}\tilde{\tau}}$ in this sum to avoid clutter, as we can easily reinstate the sum of $k$, $\ell$, $x$ and $\tilde{x}$ in the final result. Therefore, our integral becomes

$$I_{a/c} = \frac{2^{n-1}\mathcal{N}}{6i} \int_{\longrightarrow} \mathrm{d}\tau \int_{\longrightarrow} \frac{\mathrm{d}\tilde{\tau}\, \rho_{a,d,c}^{k,\ell}\, e^{-ix\tau - i\tilde{x}\tilde{\tau}}}{(-(a\tilde{\tau}+b+\tau(c\tilde{\tau}+d)))^n} \left(\tau - \frac{a\tilde{\tau}+b}{c\tilde{\tau}+d} + \frac{2a}{c} - 12s(a,c)\right) \quad (74a)$$

$$= \frac{2^{n-1}\mathcal{N} \rho_{a,d,c}^{k,\ell} e^{i\frac{xa+\tilde{x}d}{c}}}{6i} \int_{\longrightarrow} \mathrm{d}\tau \int_{\longrightarrow} \frac{\mathrm{d}\tilde{\tau}\, e^{-ix\tau - i\tilde{x}\tilde{\tau}}}{\left(\frac{1}{c} - c\tau\tilde{\tau}\right)^n} \left[\left(\tau + \frac{1}{c^2\tilde{\tau}}\right) - 12s(a,c)\right], \quad (74b)$$

where we shifted $\tau \to \tau - \frac{a}{c}$ and $\tilde{\tau} \to \tilde{\tau} - \frac{d}{c}$ in the second line. We now note the following integrals

$$\mathcal{I}_c^{(1)}(n, x, \tilde{x}) := \int_{\longrightarrow} \mathrm{d}\tau \int_{\longrightarrow} \mathrm{d}\tilde{\tau} \frac{e^{-ix\tau - i\tilde{x}\tilde{\tau}}}{\left(\frac{1}{c} - c\tau\tilde{\tau}\right)^n} = \frac{4\pi^2 (x\tilde{x})^{\frac{n-1}{2}} J_{n-1}\left(\frac{2\sqrt{x\tilde{x}}}{c}\right)}{c\, \Gamma(n)}, \quad (75)$$

$$\mathcal{I}_c^{(2)}(n, x, \tilde{x}) := \int_{\longrightarrow} \mathrm{d}\tau \int_{\longrightarrow} \mathrm{d}\tilde{\tau} \frac{e^{-ix\tau - i\tilde{x}\tilde{\tau}}}{\left(\frac{1}{c} - c\tau\tilde{\tau}\right)^n} \left(\tau + \frac{1}{c^2\tilde{\tau}}\right)$$

$$= \frac{4\pi^2 i x^{\frac{n-2}{2}} \tilde{x}^{\frac{n}{2}}}{c^2\, \Gamma(n)} \left[J_{n-2}\left(\frac{2\sqrt{x\tilde{x}}}{c}\right) - J_n\left(\frac{2\sqrt{x\tilde{x}}}{c}\right)\right], \quad (76)$$

---

[6]Since $b$ can be determined via $ad - bc = 1$, we suppress it from the notation.

whose derivation is included for completeness in appendix D. Our integral can hence succinctly be written as

$$I_{a/c} = \frac{\mathcal{N} 2^{n-1} \rho_{a,d,c}^{k,\ell} \, e^{i\frac{xa+\tilde{x}d}{c}}}{6i} \left( \mathcal{I}_c^{(2)}(n,x,\tilde{x}) - 12s(a,c)\mathcal{I}_c^{(1)}(n,x,\tilde{x}) \right), \tag{77}$$

which explicitly evaluates to[7]

$$I_{a/c} = \frac{\mathcal{N} \rho_{a,d,c}^{k,\ell} 2^n \pi^2 e^{i\frac{xa+\tilde{x}a^*}{c}}}{3c^2 \Gamma(n)} (x\tilde{x})^{\frac{n-1}{2}} \Bigg[ 12ic\, s(a,c) J_{n-1}\left(\frac{2\sqrt{x\tilde{x}}}{c}\right) \\ + \sqrt{\frac{\tilde{x}}{x}} \left( J_{n-2}\left(\frac{2\sqrt{x\tilde{x}}}{c}\right) - J_n\left(\frac{2\sqrt{x\tilde{x}}}{c}\right) \right) \Bigg]. \tag{78}$$

## 3.2 The bosonic string partition function

Let us begin by considering the example of the bosonic string partition function, $Z$, for which the form $\alpha(\tau, \tilde{\tau})$ is

$$\alpha(\tau, \tilde{\tau}) = \frac{d^2\tau}{(\operatorname{Im}\tau)^{14} |\eta(\tau)^{24}|^2}, \tag{79}$$

where $\eta(\tau)$ is the Dedekind eta function, whose relevant properties for our computations are:

$$\eta\left(\frac{a\tau + b}{c\tau + d}\right)^{24} = (c\tau + d)^{12} \eta(\tau)^{24}, \tag{80a}$$

$$\eta(\tau)^{-24} = q^{-1} + 24 + \dots, \quad \text{as} \quad \operatorname{Im}\tau \to \infty, \tag{80b}$$

where as usual $q = e^{2\pi i \tau}$. The real part of the partition function is computed by the integral

$$\operatorname{Re} Z = \oint_{\mathcal{F}} \alpha(\tau, \tilde{\tau}) = \frac{1}{6} \sum_{c=1}^{\infty} \sum_{\substack{a=0 \\ (a,c)=1}}^{c-1} I_{a/c}, \tag{81}$$

where each term is given by (2)

$$I_{a/c} = \int_{\tau \in \longrightarrow} \int_{\tilde{\tau} \in C_{a/c}} \left[ \left( \tau - \tilde{\tau} + \frac{2a}{c} \right) - 12s(a,c) \right] \alpha(\tau, \tilde{\tau}). \tag{82}$$

The computation is easily done using (78), and we find the infinite sum representation for the real part

$$\operatorname{Re} Z = \frac{(4\pi)^{15}}{24 \cdot 13!} \sum_{c=1}^{\infty} \sum_{\substack{a=0 \\ (a,c)=1}}^{c-1} \frac{e^{\frac{2\pi i(a+a^*)}{c}}}{c^2} \left[ 12ic\, s(a,c) J_{13}\left(\frac{4\pi}{c}\right) + J_{12}\left(\frac{4\pi}{c}\right) - J_{14}\left(\frac{4\pi}{c}\right) \right]. \tag{83}$$

The sum over $a$ without the inclusion of $s(a,c)$ can be recognized as the Kloosterman sum $K(1,1;c)$. Recall that the Bessel function $J_k(\frac{4\pi}{c})$ decays as $\sim c^{-k}$, which means the infinite sum converges very fast. For example, the first two terms ($c \leqslant 2$) are:

$$\operatorname{Re} Z = \frac{(4\pi)^{15}}{24 \cdot 13!} \left[ J_{12}(4\pi) - J_{14}(4\pi) + \frac{1}{4} \left( J_{12}(2\pi) + J_{14}(2\pi) \right) \right] + \dots \tag{84}$$

---

[7]It follows from the invariance of original integrand under T-modular transformations that $\rho_{a,c,d}^{k,\ell}$ transform such that the whole expression is invariant under $a^* \to a^* + c$, i.e. it is independent of the modular inverse that we choose.

These are enough to get all digits of $Z$ before the decimal point correctly. Taking $c \leqslant 1000$ gives around 30-digit accuracy:

$$\mathrm{Re}\, Z \approx 29399.071529389621479451981485159906\ldots \tag{85}$$

Since the imaginary part originates entirely from the cusp, it admits a closed form evaluation,

$$\mathrm{Im}\, Z = \frac{(4\pi)^{13}\pi}{13!} \approx 98310.0209366688109195354320180115224\ldots \tag{86}$$

Let us also note that the first line in (85) involving $s(a,c)$ is non-zero only for $c \geqslant 3$. Since the sum is very much dominated by the first few terms, the contribution from the second term in (37) is suppressed with respect to the first one by about 5 orders of magnitude in this example.

**Comparison with direction numerical integration.**    Since this is the first application of the Rademacher formula, let us contrast it with the direct numerical evaluation of $Z$. We can do it by splitting the modified fundamental domain $\mathcal{F}_{i\varepsilon}$ along some cutoff $\mathrm{Im}\, \tau = L \geqslant 1$, which divides it into $\mathcal{F}_L$, as given in (10), and its complement $\mathcal{F}_{i\varepsilon} \setminus \mathcal{F}_L$. The integral over $\mathcal{F}_L$ is convergent and hence can be evaluated numerically. In practice, it pays off to set $L = 1$ so that this region is as small as possible.

The integral over the complement $\mathcal{F}_{i\varepsilon} \setminus \mathcal{F}_L$ can be computed by first $q$-expanding the integrand and integrating each term analytically. To be concrete, we encounter integrals of the form

$$\int_{\mathcal{F}_{i\varepsilon} \setminus \mathcal{F}_L} \frac{\mathrm{d}^2\tau}{(\mathrm{Im}\, \tau)^n} q^m \bar{q}^{\tilde{m}} = \delta_{m\tilde{m}} \int_L^\infty \frac{\mathrm{d}y}{y^n} e^{-4\pi m y} = L^{1-n} E_n^*(4\pi m L). \tag{87}$$

In the first step, we plugged in $\tau = x + iy$ and integrated out the $x$ direction which fixes $m$ and $\tilde{m}$ to be the same. In the second step, we recognized that the result can be expressed in terms of the exponential integral $E_n(x)$. Our choice of the $i\varepsilon$ selects the opposite branch than the canonical choice for $E_n(x)$, which is why we included the complex conjugate above. This distinction only matters for the terms with $m < 0$. For instance, in our application we have $n = 14$ and only one exponent is negative, $m = -1$. Hence $\mathrm{Im}\, Z = \mathrm{Im}\, E_{14}^*(-4\pi) = \frac{(4\pi)^{13}\pi}{13!}$, as before. The real part can be similarly computed to high accuracy.

To summarize, the bottleneck in the direct numerical evaluation is in computing the integral over $\mathcal{F}_L$. In the Mathematica notebook attached as an ancillary file, we have implemented this procedure and confirmed (85) to 20 digits using similar computational time spent on getting the 30 digits using the Rademacher formula. Direct integration is clearly subpar. We also checked that the result agrees with the value provided in [3, 37, 41] to the number of digits specified there (or exactly for the imaginary part).

**Relation to the cosmological constant.**    We can relate the above evaluation of the bosonic string partition function to the one-loop cosmological constant by matching with the vacuum transition amplitude. The vacuum transition amplitude in the target spacetime theory is a sum of vacuum bubble diagrams which in this case is a sum of all the multi-torus closed string partition functions. This sum exponentiates and so we have

$$\langle 0 | e^{-iHT} | 0 \rangle = e^{-i\Lambda V_{26}} = e^{Z_{\mathrm{T}^2}}, \tag{88}$$

where $\Lambda$ is the cosmological constant or vacuum energy density and $V_{26}$ is the volume of the 26-dimensional Lorentzian target spacetime. The torus-partition function is given by

$$Z_{\mathrm{T}^2} = \frac{iV_{26}}{2(2\pi\ell_s)^{26}} \int_{\mathcal{F}_{i\varepsilon}} \frac{\mathrm{d}^2\tau}{(\mathrm{Im}\, \tau)^{14} |\eta(\tau)^{24}|^2}. \tag{89}$$

We thus have an expression for the cosmological constant [42],

$$\Lambda = -\frac{1}{2(2\pi\ell_s)^{26}} \int_{\mathcal{F}_{i\varepsilon}} \frac{\mathrm{d}^2\tau}{(\operatorname{Im}\tau)^{14} |\eta(\tau)^{24}|^2}. \tag{90}$$

We observe that $\Lambda$ is complex with imaginary part given by

$$\operatorname{Im}\Lambda = -\frac{1}{2\pi^{12} \cdot 13! \ell_s^{26}}. \tag{91}$$

The fact that $\operatorname{Im}\Lambda < 0$ means that the vacuum amplitude decays which is indicative of instability. The real part follows directly from (85).

**Generalization to $n$ bosons.** We can readily generalize the result to express the integral of the partition function of $n \leqslant 24$ free bosons over the fundamental domain denoted $Z_n$ as

$$\begin{aligned}
\operatorname{Re} Z_n = {}& \frac{4\pi^2}{3} \frac{\left(\frac{\pi n}{6}\right)^{\frac{n}{2}}}{\Gamma\left(\frac{n}{2}+2\right)} \\
& \times \sum_{c=1}^{\infty} \sum_{\substack{a=0\\(a,c)=1}}^{c-1} \frac{e^{\pi i n s(a,c)}}{c^2} \left[ \pi i n c \, s(a,c) J_{\frac{n}{2}+1} + \frac{\pi n}{12} \left( J_{\frac{n}{2}}\left(\frac{\pi n}{6c}\right) - J_{\frac{n}{2}+2}\left(\frac{\pi n}{6c}\right) \right) \right].
\end{aligned} \tag{92}$$

The Dedekind sum now also appears in the exponent due to the multiplier phase in the transformation formula of the Dedekind eta function (51). The imaginary part coming entirely from the Euclidean cusp evaluates to

$$\operatorname{Im} Z_n = \frac{\left(\frac{\pi n}{6}\right)^{\frac{n}{2}+1} \pi}{\Gamma\left(\frac{n}{2}+2\right)}. \tag{93}$$

For $n > 24$ bosons, there are more polar terms that contribute to the integral over the Ford circles.

## 3.3 First mass shift of type II strings

The next application is to computing the leading mass shift $\delta m^2$ of the level-1 states in type-II superstring, which allows us to compute the renormalized mass, $m^2 = \frac{1}{\alpha'}(1 + \delta m^2 + \dots)$. The mass shift is complex with $\operatorname{Im}\delta m^2$ measuring the decay width of the level-1 states. It can be extracted from the real part of a $1 \to 1$ scattering amplitude for level-1 vertex operators at genus one, which is given by an integral over the moduli space $\mathcal{M}_{1,2}$ of a torus with 2 punctures. In our normalizations, a concrete expression is

$$\delta m^2 = \frac{1}{16\pi^2} \int_{\mathcal{M}_{1,2}} \frac{\mathrm{d}^2\tau \, \mathrm{d}^2 z}{(\operatorname{Im}\tau)^5} \left| \frac{\vartheta_1(z|\tau)^2}{\eta(\tau)^6} \right|^2 e^{-\frac{4\pi(\operatorname{Im}z)^2}{\operatorname{Im}\tau}}. \tag{94}$$

The $\tau$-integral is over the (modified) fundamental domain $\mathcal{F}$, while the $z$-integral is over the once-punctured torus $\mathbb{T} \setminus \{0\}$ since we can fix the position of one puncture to the origin. The $i\varepsilon$ prescription described in section 2.6.3 is needed.

Thus, after integrating out $z$, which can be done analytically [43], we have a leftover $\tau$-integral which can be done using our two-dimensional Rademacher contour. Integrating out $z$ leads to the integrand

$$\alpha(\tau, \tilde{\tau}) = \frac{1}{32\pi^2} \frac{\mathrm{d}^2\tau}{(\operatorname{Im}\tau)^{\frac{9}{2}}} \frac{|\vartheta_3(2\tau)|^2 + |\vartheta_2(2\tau)|^2}{|\eta(\tau)^6|^2}. \tag{95}$$

We now encounter non-trivial multiplier phases in the modular transformation of the integrand. Let us define

$$f(\tau) = \frac{1}{\eta(\tau)^6} \begin{pmatrix} \vartheta_2(2\tau) \\ \vartheta_3(2\tau) \end{pmatrix}. \tag{96}$$

Then $f$ is a vector-valued modular form of weight $-\frac{5}{2}$ with transformation law for $c > 0$

$$f\left(\frac{a\tau + b}{c\tau + d}\right) = \frac{1}{(-i(c\tau + d))^{\frac{5}{2}}} \rho_{a,d,c} f(\tau), \tag{97}$$

with

$$\rho_{a,d,c} = \frac{1}{\sqrt{2c}} \begin{pmatrix} -G_0(d,c) & G_1(d,c) \\ i^{ab} G_a(d,c) & -i^{(a+2)b} G_{a+1}(d,c) \end{pmatrix}, \tag{98}$$

where we defined the shifted Gauss sum

$$G_a(d,c) = \sum_{k \in \mathbb{Z}_c + \frac{a}{2}} e^{-\frac{2\pi i d k^2}{c}}. \tag{99}$$

We included a derivation of this fact for completeness in appendix E. We can then write

$$\alpha(\tau, \tilde{\tau}) = \frac{1}{32\pi^2} \frac{d^2 \tau}{(\operatorname{Im} \tau)^{\frac{9}{2}}} f(\tau)^{\mathsf{T}} f(\tilde{\tau}). \tag{100}$$

Using (78) and noting that only $\frac{\vartheta_3(2\tau)}{\eta(\tau)^6}$ has a single polar term, we get

$$I_{a/c} = -i^{(a+2)b} \frac{\sqrt{2}\pi^3 e^{\frac{i\pi(a+d)}{2c}} G_{a+1}(d,c)}{630 c^{\frac{5}{2}}} \left[ 12 i c \, s(a,c) J_{\frac{7}{2}}\left(\frac{\pi}{c}\right) + J_{\frac{5}{2}}\left(\frac{\pi}{c}\right) - J_{\frac{9}{2}}\left(\frac{\pi}{c}\right) \right]. \tag{101}$$

We can alternatively apply the transformation as in (E.12) backwards to get rid of the prefactor and write it in terms of a Gauss sum. Renaming also $d \to a$ leads to

$$\operatorname{Re} \delta m^2 = -\frac{\sqrt{2}\pi^3}{630} \sum_{c=1}^{\infty} \frac{1}{c^{\frac{5}{2}}} \sum_{\substack{a=0 \\ (a,c)=1}}^{c-1} \sum_{m=0}^{c-1} e^{-\frac{\pi i}{c}(2am^2 + 2(a-1)m - 1)}$$

$$\times \left[ 12 i c \, s(a,c) J_{\frac{7}{2}}\left(\frac{\pi}{c}\right) + J_{\frac{5}{2}}\left(\frac{\pi}{c}\right) - J_{\frac{9}{2}}\left(\frac{\pi}{c}\right) \right]. \tag{102}$$

The first few digits are:

$$\operatorname{Re} \delta m^2 \approx 0.02204255\ldots \tag{103}$$

The imaginary part has an exact formula equal to

$$\operatorname{Im} \delta m^2 = \frac{\pi^2}{210} \approx 0.04699812\ldots \tag{104}$$

One can again readily compare this formula with the direct numerical integration over the fundamental domain. For this, it is useful to evaluate the Gauss sums in closed form in terms of Jacobi symbols as is explained in appendix E.3. Implementation of both the Rademacher formula and the direct numerical integration are provided in the ancillary Mathematica notebook.

## 3.4 Toroidal compactification of the bosonic string

As another example, let us evaluate the torus partition function of the closed bosonic string when one of the target spacetime directions has been compactified into a circle of radius $R$. This is the integral over the fundamental domain of the partition function of 23 non-compact bosons and 1 compact boson of radius $R$,

$$Z(\tau, \tilde{\tau}) = \frac{1}{(\mathrm{Im}\,\tau)^{\frac{23}{2}} |\eta(\tau)|^{48}} \sum_{(m,w) \in \mathbb{Z} \times \mathbb{Z}} q^{\frac{1}{4}\left(\frac{m}{R} + wR\right)^2} \tilde{q}^{\frac{1}{4}\left(\frac{m}{R} - wR\right)^2}, \tag{105}$$

where $m$ and $w$ are respectively the momentum and winding numbers of the string around the compact direction. Only primary states satisfying the inequality

$$-2 < \frac{m}{R} + wR < 2, \tag{106}$$

contribute since they are singular at the cusp $\tau = i\infty$. For simplicity, we work at the self-dual radius $R = 1$ in which case there are three families of solutions to the inequality given by

$$m + w = 0, \pm 1. \tag{107}$$

The states with $m + w = 0$ have $h = 0$ and those with $m + w = \pm 1$ have $h = \frac{1}{4}$ and appear with a multiplicity of 2. This example is actually quite similar to the computation of the mass-shift, since the self-dual free boson partition function is given by

$$Z_{R=1}(\tau, \tilde{\tau}) = \frac{|\vartheta_3(2\tau)|^2 + |\vartheta_2(2\tau)|^2}{|\eta(\tau)|^2}, \tag{108}$$

with the two terms corresponding to the two characters when we interpret the theory as the $\mathfrak{su}(2)_1$ WZW model. Thus the mass-shift computation was the case where we integrated the partition function of self-dual free boson together with five non-compact free bosons. Since it is no more complicated, we will write the integral of $n - 1$ free bosons with $n \leq 24$ and one self-dual free boson, which gives a thermal version of (92).

For $6 \leq n \leq 24$, the four entries of $\rho^{k,\ell}_{a,d,c} M^k(x) \tilde{M}^\ell(\tilde{x})$ have $(3-k)(3-\ell)$ polar terms, all with the same values of $(x, \tilde{x})$. The $k = 1$ entry corresponds to the $h = \frac{1}{4}$ states and the $k = 2$ entry corresponds to the $h = 0$ states that we discussed above. The total multiplicity of the polar terms is the multiplicity of the left- and right-moving multiplicity. Let us call $x_1 = \frac{\pi(n-6)}{12}$, $x_2 = \frac{\pi n}{12}$. Then the values of $(x, \tilde{x})$ in (73) appearing in $\rho^{k,\ell}_{a,d,c} M^k(x) \tilde{M}^\ell(\tilde{x})$ are $(x, \tilde{x}) = (x_k, x_\ell)$. Let us continue to denote (98) by $\rho_{a,c,d}$. Then[8]

$$I_{a/c} = \sum_{k,\ell=1}^{2} \frac{2^{\frac{n+3}{2}} \pi^2 (3-k)(3-\ell)(x_k x_\ell)^{\frac{n+1}{4}}}{3c^2 \Gamma\left(\frac{n+3}{2}\right)} e^{\frac{\pi i}{2c}(a(k-1)+d(\ell-1))+\pi i(n-6)s(a,c)} \rho^{k,\ell}_{a,d,c}$$

$$\times \left[ 12 i c\, s(a,c) J_{\frac{n+1}{2}}\left(\frac{2\sqrt{x_k x_\ell}}{c}\right) + \sqrt{\frac{x_\ell}{x_k}} \left( J_{\frac{n-1}{2}}\left(\frac{2\sqrt{x_k x_\ell}}{c}\right) - J_{\frac{n+3}{2}}\left(\frac{2\sqrt{x_k x_\ell}}{c}\right) \right) \right]. \tag{109}$$

In particular, it is straightforward to evaluate this for $n = 24$ with the result

$$\mathrm{Re}\, Z = 31928.3317550443028896792811896926852 \ldots, \tag{110}$$

which can also be verified by direct numerical integration. The imaginary part can be computed exactly for any $R$ and takes the form

$$\mathrm{Im}\, Z = \frac{(4\pi)^{\frac{25}{2}} \pi}{\Gamma\left(\frac{27}{2}\right)} \left[ 1 + 2 \sum_{m=1}^{\lfloor 2R^{-1} \rfloor} \left(1 - \frac{m^2 R^2}{4}\right)^{\frac{25}{2}} + 2 \sum_{m=1}^{\lfloor 2R \rfloor} \left(1 - \frac{m^2}{4R^2}\right)^{\frac{25}{2}} \right]. \tag{111}$$

---

[8] This formula is also correct for $n \leq 6$, except that one only includes $k = \ell = 2$ in the sum for that case.

## 3.5 SO(16) × SO(16) string

The simplest modular integrals that one encounters in a string theory context are the integrals computing the one-loop partition function. We computed this quantity for the bosonic string partition function in section 3.2 and the thermal bosonic string partition function in section 3.4. The instability of the theory was reflected in the imaginary part of the partition function. For supersymmetric string theories on the other hand, the one-loop partition function vanishes. There are however a number of tachyon-free non-supersymmetric strings which have a finite real one-loop partition function. The most well-known instance is the $SO(16) \times SO(16)$ string in ten dimensions, see also [44] for an overview of many more constructions of non-supersymmetric strings in diverse dimensions. This is a version of the heterotic string. It can be constructed by coupling the spin structures of the 8 right-moving fermions parametrizing the target space dimensions to the spin structure of the fermions obtained by fermionizing the internal dimensions of the heterotic string in a particular way [19, 20]. The one-loop cosmological constant is given by $\Lambda = -\frac{1}{2(2\pi\ell_s)^{10}} \int_{\mathcal{F}_{i\varepsilon}} \omega(\tau, \tilde{\tau})$ with

$$\omega(\tau, \tilde{\tau}) = \frac{\mathrm{d}^2\tau}{(\mathrm{Im}\,\tau)^6} \frac{1}{\eta(\tau)^{12}} \left( \frac{\vartheta_2^4(\tau)}{\vartheta_2^8(\tilde{\tau})} + \frac{\vartheta_4^4(\tau)}{\vartheta_4^8(\tilde{\tau})} - \frac{\vartheta_3^4(\tau)}{\vartheta_3^8(\tilde{\tau})} \right). \tag{112}$$

Since the integrand doesn't have a tachyon divergence as $\tau, \tilde{\tau} \to i\infty$, the imaginary part of the integral vanishes and we can directly use our integration formula for $f$. Let

$$f_{\binom{1}{0}}(\tau) = \frac{\vartheta_2(\tau)^4}{\eta(\tau)^{12}}, \qquad f_{\binom{1}{1}}(\tau) = -\frac{\vartheta_3(\tau)^4}{\eta(\tau)^{12}}, \qquad f_{\binom{0}{1}}(\tau) = \frac{\vartheta_4(\tau)^4}{\eta(\tau)^{12}}, \tag{113a}$$

$$g_{\binom{1}{0}}(\tau) = \frac{1}{\vartheta_2(\tau)^8}, \qquad g_{\binom{1}{1}}(\tau) = \frac{1}{\vartheta_3(\tau)^8}, \qquad g_{\binom{0}{1}}(\tau) = \frac{1}{\vartheta_4(\tau)^8}. \tag{113b}$$

The subscripts are to be read mod 2. Then

$$f_{\left(\begin{smallmatrix} a & b \\ c & d \end{smallmatrix}\right)\binom{\alpha}{\beta}}\left(\frac{a\tau + b}{c\tau + d}\right) = (c\tau + d)^{-4} f_{\binom{\alpha}{\beta}}(\tau), \tag{114a}$$

$$g_{\left(\begin{smallmatrix} a & b \\ c & d \end{smallmatrix}\right)\binom{\alpha}{\beta}}\left(\frac{a\tau + b}{c\tau + d}\right) = (c\tau + d)^{-4} g_{\binom{\alpha}{\beta}}(\tau). \tag{114b}$$

One can verify this easily on the generators for the S- and T-modular transformations. Since this transformation respects the $SL(2,\mathbb{Z})$ group law, it follows for all $SL(2,\mathbb{Z})$ transformations.

We can thus package $f_{\binom{\alpha}{\beta}}(\tau)$ and $g_{\binom{\alpha}{\beta}}(\tau)$ into the following vector-valued modular forms:

$$f(\tau) = \begin{pmatrix} f_{\binom{1}{0}}(\tau) \\ f_{\binom{1}{1}}(\tau) \\ f_{\binom{0}{1}}(\tau) \end{pmatrix}, \qquad g(\tau) = \begin{pmatrix} g_{\binom{1}{0}}(\tau) \\ g_{\binom{1}{1}}(\tau) \\ g_{\binom{0}{1}}(\tau) \end{pmatrix}. \tag{115}$$

Since this integrand doesn't respect the parity assumption (8), we consider

$$\alpha(\tau, \tilde{\tau}) = \frac{1}{2}(\omega(\tau, \tilde{\tau}) - \omega(\tilde{\tau}, \tau)) = \frac{1}{2} \frac{\mathrm{d}^2\tau}{\mathrm{Im}(\tau)^6} \left( f(\tau)^{\mathsf{T}} g(\tilde{\tau}) + g(\tau)^{\mathsf{T}} f(\tilde{\tau}) \right). \tag{116}$$

We have to determine the polar terms that contribute to the integral as in eq. (73). We observe that, near the cusp, the only modular forms with polar terms are

$$f_{\binom{1}{1}}(\tau) = -\mathrm{e}^{-\pi i \tau} + \dots, \quad f_{\binom{0}{1}}(\tau) = \mathrm{e}^{-\pi i \tau} + \dots, \quad g_{\binom{1}{0}}(\tau) = 2^{-8}\mathrm{e}^{-2\pi i \tau} + \dots \tag{117}$$

For the $f(\tau)^{\mathsf{T}} g(\tilde{\tau})$ term, we hence have

$$f(\tau)^{\mathsf{T}} \rho_{a,d,c} g(\tilde{\tau}) = \sum_{(\alpha,\beta)=(1,0),(0,1),(1,1)} f_{\binom{\alpha}{\beta}}(\tau) g_{\left(\begin{smallmatrix} d & b \\ c & a \end{smallmatrix}\right)\binom{\alpha}{\beta}}(\tilde{\tau}), \tag{118}$$

$f$ only contributes for $\beta \equiv 1 \bmod 2$, in which case it will provide the coefficient $(-1)^{\alpha}$. $g$ contributes only for $\left(\begin{smallmatrix} d & b \\ c & a \end{smallmatrix}\right)\binom{\alpha}{\beta} \equiv \binom{1}{0} \bmod 2$, i.e. $\binom{\alpha}{\beta} \equiv \left(\begin{smallmatrix} a & b \\ c & d \end{smallmatrix}\right)\binom{1}{0} \equiv \binom{a}{c} \bmod 2$. Thus, for both terms to contribute, $c$ has to be odd and we get a sign $(-1)^{\alpha} = (-1)^{a}$. Thus the replacement (73) reads in this case

$$f(\tau)^{\mathsf{T}} g(\tilde{\tau}) \to 2^{-8}(-1)^{a} \delta_{c \in 2\mathbb{Z}+1} e^{-i\pi\tau - 2\pi i \tilde{\tau}}. \tag{119}$$

As for the $g(\tau)^{\mathsf{T}} f(\tilde{\tau})$ term, the cusp expansion is

$$g(\tau)^{\mathsf{T}} \rho_{a,d,c} f(\tilde{\tau}) = \sum_{(\alpha,\beta)=(1,0),(0,1),(1,1)} g_{\binom{\alpha}{\beta}}(\tau) f_{\left(\begin{smallmatrix} d & b \\ c & a \end{smallmatrix}\right)\binom{\alpha}{\beta}}(\tilde{\tau}) \tag{120a}$$

$$= \sum_{(\alpha,\beta)=(1,0),(0,1),(1,1)} f_{\binom{\alpha}{\beta}}(\tilde{\tau}) g_{\left(\begin{smallmatrix} a & b \\ c & d \end{smallmatrix}\right)\binom{\alpha}{\beta}}(\tau), \tag{120b}$$

where we renamed $\binom{\alpha}{\beta} \to \left(\begin{smallmatrix} a & b \\ c & d \end{smallmatrix}\right)\binom{\alpha}{\beta}$ in the second line. Thus this case is related to the previous one by exchanging $a \leftrightarrow d$ and $\tau \leftrightarrow \tilde{\tau}$ and hence the replacement rule (73) becomes

$$g(\tau)^{\mathsf{T}} f(\tilde{\tau}) \to 2^{-8}(-1)^{d} \delta_{c \in 2\mathbb{Z}+1} e^{-2i\pi\tau - \pi i \tilde{\tau}}. \tag{121}$$

Therefore with the help of (69), our integral $I_{a/c}$ reads

$$I_{a/c} = \frac{\pi^7 \delta_{c \in 2\mathbb{Z}+1} e^{\frac{i\pi(a+d)}{c}}}{720 c^2} \left[ 12 i \sqrt{2} c J_5\left(\frac{2\sqrt{2}\pi}{c}\right)\left((-1)^d e^{\frac{i\pi a}{c}} + (-1)^a e^{\frac{i\pi d}{c}}\right) s(a,c) \right.$$
$$\left. + \left(J_4\left(\frac{2\sqrt{2}\pi}{c}\right) - J_6\left(\frac{2\sqrt{2}\pi}{c}\right)\right)\left((-1)^d e^{\frac{i\pi a}{c}} + 2(-1)^a e^{\frac{i\pi d}{c}}\right) \right]. \tag{122}$$

By considering the full sum (2), we can obtain a simpler form. Since we sum over all $a$'s, we can rename $d = a^* \to a$ in in the terms originating from $g(\tau)^{\mathsf{T}} f(\tilde{\tau})$ in (122) and write

$$I = \sum_{\substack{c \text{ odd}}} \sum_{\substack{a=0 \\ (a,c)=1}}^{c-1} \frac{\pi^7 e^{\frac{\pi i (2a+(c+1)a^*)}{c}}}{240 c^2} \left[ 8 i \sqrt{2} c\, s(a,c) J_5\left(\frac{2\sqrt{2}\pi}{c}\right) + J_4\left(\frac{2\sqrt{2}\pi}{c}\right) - J_6\left(\frac{2\sqrt{2}\pi}{c}\right) \right]. \tag{123}$$

Notice that the combination $e^{\frac{\pi i (2a+a^*)}{c}}(-1)^{a^*} = e^{\frac{\pi i (2a+(c+1)a^*)}{c}}$ that appears in the prefactor is precisely independent of the choice of $a^*$ when $c$ is odd. This formula gives the approximation

$$I \approx -5.6716185652722887166\ldots \tag{124}$$

Due to the additional sign between the one-loop cosmological constant $\Lambda$ and $I$, this confirms that $\Lambda > 0$ in the ten-dimensional $SO(16) \times SO(16)$ string.

## 3.6 Rational CFT partition function

We will consider another interesting class of integrals of the form

$$\alpha(\tau, \tilde{\tau}) = \frac{\mathrm{d}^2\tau}{(\operatorname{Im}\tau)^2} Z_{\mathrm{CFT}}(\tau, \bar{\tau} = -\tilde{\tau}), \tag{125}$$

for a unitary 2d CFT partition function. In order to have more control over the computation, we will consider a rational CFT partition function with a diagonal modular invariant. This leads to an integrand of the form (69) with $\mathcal{N} = 1$, $n = 2$ and $f = g$. The vector-valued modular form $f(\tau)$ is the vector of characters of the CFT.

These examples are interesting from a theoretical standpoint because they violate the convergence condition (B.1). Thus we are in principle not guaranteed to obtain sensible answers. We will however see in two examples that our formula still works (with small modifications).

### 3.6.1 Compact free boson

The partition function of a compact boson of radius $R$ is given by

$$Z(\tau, \tilde{\tau}) = \frac{1}{|\eta(\tau)|^2} \sum_{(m,w) \in \mathbb{Z} \times \mathbb{Z}} q^{\frac{1}{4}\left(\frac{m}{R} + wR\right)^2} \tilde{q}^{\frac{1}{4}\left(\frac{m}{R} - wR\right)^2}, \tag{126}$$

where $m$ and $w$ are the momentum and winding numbers which label the primaries in the spectrum as in (105). At the self-dual radius $R = 1$, we already computed the Rademacher expansion of the integral over $Z(\tau, \tilde{\tau})$, which is given by (109) for $n = 1$. As remarked in the footnote there, we only have to keep the $k = \ell = 2$ in the sum and obtain

$$I_{a/c} = \frac{\pi^3}{9c^2} e^{\frac{\pi i}{2c}(a+d) - 5\pi i s(a,c)} \rho_{a,d,c}^{2,2} \left[ 12ic\, s(a,c) J_1\left(\frac{\pi}{6c}\right) + J_0\left(\frac{\pi}{6c}\right) - J_2\left(\frac{\pi}{6c}\right) \right] \tag{127a}$$

$$= -\frac{\pi^3}{9\sqrt{2}c^{\frac{5}{2}}} e^{-5\pi i s(a,c)} \sum_{m=0}^{c-1} e^{-\frac{\pi i}{c}(2dm^2 + 2(d-1)m - 1)}$$

$$\times \left[ 12ic\, s(a,c) J_1\left(\frac{\pi}{6c}\right) + J_0\left(\frac{\pi}{6c}\right) - J_2\left(\frac{\pi}{6c}\right) \right]. \tag{127b}$$

As already noticed, the integrand $\frac{d^2\tau}{(\text{Im}\,\tau)^2} Z_{R=1}(\tau, \tilde{\tau})$ fails the convergence criterion (B.1) and correspondingly the Rademacher sum

$$\sum_{c=1}^{\infty} \sum_{\substack{a=0 \\ (a,c)=1}}^{c-1} I_{a/c}, \tag{128}$$

is not absolutely convergent. However, it still seems to be conditionally convergent and we can pose the obvious question whether it converges to the integral $\int_{\mathcal{F}} \frac{d^2\tau}{(\text{Im}\,\tau)^2} Z_{R=1}(\tau, \tilde{\tau})$. While we haven't tried to answer this question analytically, the numerical evidence for this is strongly affirmative. In Figure 7, we plotted the difference of the Rademacher sum and the numerical integral as a function of the upper cutoff $c_{\text{max}}$ in the Rademacher sum. While convergence is slow, the figure clearly indicates that the sum converges to the correct value.

Note that this calculation shows that the integral of the partition function over the fundamental domain at the self-dual radius only gets contribution from the primary states with $h = 0$. At any given value of the radius $R$, the terms singular at the cusp $\tau \to i\infty$ are those which solve the inequality,

$$-\frac{1}{\sqrt{6}} < \frac{m}{R} + wR < \frac{1}{\sqrt{6}}. \tag{129}$$

Another value of the radius where only $h = 0$ primaries contribute is $R = 2$ which corresponds to the integral of the partition function of the $\mathbb{Z}_2$-orbifold of the compact boson at $R = 1$. There are non-vacuum primaries with $h = 0$ only if $R^2$ takes a rational value i.e., $R^2 = \frac{p}{q}$ with $(p,q) = 1$. In this case, only $h = 0$ primaries solve the inequality (129) and hence contribute to the Rademacher expansion provided $pq \leqslant 6$ corresponding to the seven values of the radius, $R^2 = \{1, \frac{3}{2}, 2, 3, 4, 5, 6\}$. If $pq > 6$, there are primary states with non-zero $h$ whose momentum and winding numbers obey the condition $mq + wp = 1$ that solves the inequality in (129) and hence contribute to the Rademacher expansion.

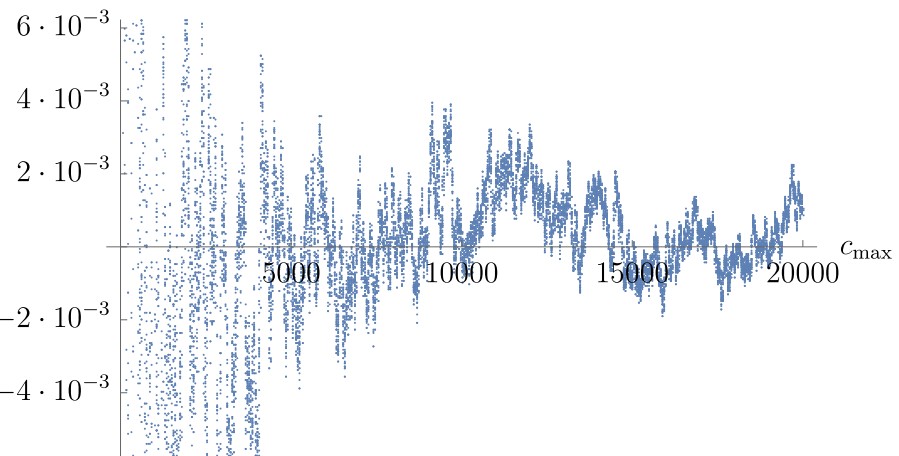

Figure 7: The difference $\sum_{c=1}^{c_{max}} \sum_{a=0,(a,c)=1}^{c-1} I_{a/c} - \oint \frac{\mathrm{d}^2\tau}{(\mathrm{Im}\,\tau)^2} Z_{R=1}(\tau, \tilde\tau)$. The error in the plot behaves roughly as $c_{max}^{-1}$. For reference, the integral evaluates numerically to $\oint \frac{\mathrm{d}^2\tau}{(\mathrm{Im}\,\tau)^2} Z_{R=1}(\tau, \tilde\tau) = 1.5708311090031060\ldots$ and hence the relative error for $c_{max} = 20000$ is about $10^{-3}$.

### 3.6.2 Monster CFT

Let us discuss as a second example the non-holomorphic Monster CFT, for which the partition function factorizes. The integrand is of the form

$$\alpha(\tau, \tilde\tau) = \frac{\mathrm{d}^2\tau}{(\mathrm{Im}\,\tau)^2} j(\tau) j(\tilde\tau). \tag{130}$$

The holomorphic $j$-function is uniquely determined from the two conditions

$$j\left(\frac{a\tau + b}{c\tau + d}\right) = j(\tau), \tag{131a}$$

$$j(\tau) = q^{-1} + 196884q + \ldots, \quad \text{as} \quad \mathrm{Im}\,\tau \to \infty. \tag{131b}$$

We can immediately evaluate the right hand side of the Rademacher formula (2) and find with the help of (78) the contribution from the Ford circle $C_{a/c}$

$$I_{a/c} = \frac{8\pi^3 \mathrm{e}^{\frac{2\pi i(a+a^*)}{c}}}{3c^2} \left[12ic\,s(a,c)J_1\left(\frac{4\pi}{c}\right) + J_0\left(\frac{4\pi}{c}\right) - J_2\left(\frac{4\pi}{c}\right)\right]. \tag{132}$$

As before, the sum over $I_{a/c}$ only converges conditionally. However, while we found numerically that it converges to the correct value for the self-dual boson, this is no longer the case in the present case. To understand the reason for this, let us first notice that we could have added to the integrand a linear combination of the terms

$$\frac{\mathrm{d}^2\tau}{(\mathrm{Im}\,\tau)^2}\left(j(\tau) + j(\tilde\tau)\right), \quad \text{or} \quad \frac{\mathrm{d}^2\tau}{(\mathrm{Im}\,\tau)^2}. \tag{133}$$

Both of these terms would lead to identical Rademacher expansions and thus we can at best hope that

$$\sum_{\substack{c=1 \\ }}^{\infty} \sum_{\substack{a=0 \\ (a,c)=1}}^{c-1} I_{a/c} \overset{?}{=} \oint \frac{\mathrm{d}^2\tau}{(\mathrm{Im}\,\tau)^2}\left(j(\tau)j(\tilde\tau) + A(j(\tau) + j(\tilde\tau)) + B\right) \tag{134a}$$

$$= \oint \frac{\mathrm{d}^2\tau}{(\mathrm{Im}\,\tau)^2} j(\tau)j(\tilde\tau) - 16\pi A + \frac{\pi B}{3}, \tag{134b}$$

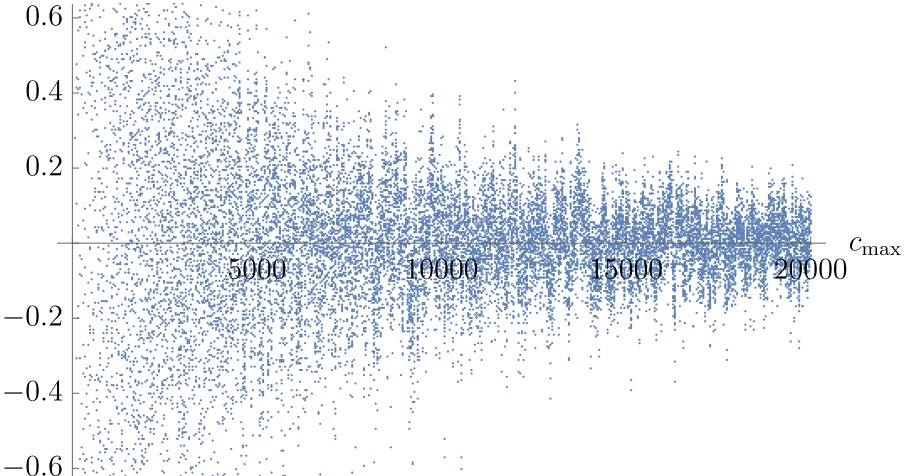

Figure 8: The difference $\sum_{c=1}^{c_{\max}} \sum_{a=0,(a,c)=1}^{c-1} I_{a/c} - \oint \frac{d^2\tau}{(\operatorname{Im}\tau)^2} j(\tau)j(\tilde{\tau}) + 384\pi$. The error in the plot behaves roughly as $c_{\max}^{-1}$. For reference, the numerical integral evaluates to $\oint \frac{d^2\tau}{(\operatorname{Im}\tau)^2} j(\tau)j(\tilde{\tau}) = 808.6857685243\ldots$

for some choice of $A$ and $B$.[9] By numerically evaluating the integrals and the Rademacher sum, we find good evidence that

$$\sum_{\substack{c=1 \\ (a,c)=1}}^{\infty} \sum_{a=0}^{c-1} I_{a/c} \overset{!}{=} \oint \frac{d^2\tau}{(\operatorname{Im}\tau)^2} j(\tau)j(\tilde{\tau}) - 384\pi, \tag{136}$$

which suggests $A = 24$ and $B = 0$. In Figure 8, we plot the difference of the LHS and the RHS of (136) as a function of the cutoff $c \leqslant c_{\max}$ that we employ to compute the LHS.

This resonates with the holomorphic counterpart of this computation [46], where the Fourier coefficients $j(\tau) = q^{-1} + \sum_{n=0}^{\infty} c_n q^n$ can be computed for $n \geqslant 1$ via the formula

$$c_n = \frac{2\pi}{\sqrt{n}} \sum_{\substack{c=1 \\ (a,c)=1}}^{\infty} \sum_{a=0}^{c-1} e^{2\pi i \frac{na-a^*}{c}} I_1\left(\frac{4\pi\sqrt{n}}{c}\right). \tag{137}$$

One encounters a similar ambiguity as above for $n = 0$ since we can shift $j(\tau) \to j(\tau) + C$ without changing its modular properties. However one can take the $n \to 0$ limit of (137) without problems and finds $c_0 = 24$. Thus the Rademacher procedure seems to prefer 24 as a constant coefficient in the Fourier expansion. Not surprisingly, this matches the constant that we found in (136), since the derivation of that formula involves the Rademacher expansion of the holomorphic $j$-function.

In general, we thus expect that convergence of the Rademacher formula of (125) becomes problematic when there are states with twist exactly $\frac{c}{24}$, since they are invisible to the Rademacher procedure. In the free boson case, no such states exist which gives at least an intuitive reason why convergence continued to hold in that case.

---

[9]The integral involving the holomorphic $j$-function can be done analytically using the technique described in [45]. In our conventions, one has for a holomorphic modular invariant function

$$\oint_{\mathcal{F}} \frac{d^2\tau}{(\operatorname{Im}\tau)^2} F(\tau) = \frac{\pi}{3} E_2(\tau) F(\tau)\big|_{q^0}, \tag{135}$$

where $E_2(\tau) = 1 - 24\sum_{n=1}^{\infty} \frac{nq^n}{1-q^n}$ is the second Eisenstein series and we pick out the $q^0$ term of the Fourier series.

## Acknowledgments

We thank Nathan Benjamin, Scott Collier, Lance Dixon, Abhiram Kidambi, Alex Maloney, Jan Manschot, Sridip Pal, Ioannis Tsiares and Nicolas Valdes-Meller for useful discussions.

**Funding information**  MMB and LE are supported by the European Research Council (ERC) under the European Union's Horizon 2020 research and innovation programme (grant agreement No 101115511). JC and TH are supported by NSF grant PHY-2309456.

## A Counting Ford circles

Let us give the proof for the recursion relation (44) for completeness. Let us recall the claim. We set for $x \in \mathbb{Q} \cap [0, 1)$,

$$m(x) = \# \left\{ y \in \mathbb{Q}_{>0}, g \in \text{PSL}(2, \mathbb{Z}) \mid x = g \cdot y, \ g^{-1} \cdot \infty < -1 \right\}, \qquad \text{(A.1)}$$

where $\cdot$ denotes the usual action of $\text{PSL}(2, \mathbb{Z})$ on $\mathbb{Q} \cup \{\infty\}$. Then $m$ can be computed recursively in the size of the denominator $\gamma$ from the recursion relation

$$m(x) = \begin{cases} m\left(\frac{x}{1-x}\right), & x < \frac{1}{2}, \\ m\left(\frac{2x-1}{x}\right) + 1, & x > \frac{1}{2}, \end{cases} \qquad \text{(A.2)}$$

with initial condition $m(0) = m(\frac{1}{2}) = 0$.

We prove this by induction in the size of the denominator. Hence we first check the two initial cases and then consider the two cases in (A.2) separately.

To bring this into the form used in the main text, we set (with $c > 0$, $\gamma > 0$ and $(a, c) = 1$, $(\alpha, \gamma) = 1$)

$$x = \frac{\alpha}{\gamma}, \qquad g = \begin{pmatrix} \tilde{d} & \tilde{b} \\ \tilde{c} & \tilde{a} \end{pmatrix}, \qquad y = \frac{a}{c}. \qquad \text{(A.3)}$$

The condition $g^{-1} \cdot \infty < -1$ translates to $\frac{\tilde{a}}{\tilde{c}} > 1$. Notice in particular that these conditions imply

$$c\tilde{c} < c\tilde{c} \left( \frac{a}{c} + \frac{\tilde{a}}{\tilde{c}} \right) = \gamma. \qquad \text{(A.4)}$$

Thus there are only finitely many solutions for $c$ and $\tilde{c}$ and the condition $\tilde{a}c + a\tilde{c} = \gamma$ gives finitely many solutions for $a$ and $\tilde{c}$ for given $c$ and $\tilde{c}$. Thus $m(\frac{\alpha}{\gamma})$ is finite.

### A.1 Initial conditions

Consider first $m(0)$. Since $\gamma = 1$, the bound (A.4) has no solution and thus $m(0) = 0$.

For $m(\frac{1}{2})$, the only solution to the bound (A.4) is $c = \tilde{c} = 1$. We then have $a \geqslant 1$ and $\tilde{a} \geqslant 2$. But this contradicts $\tilde{a}c + a\tilde{c} = \gamma = 2$ and so there is no solution.

### A.2 Recursion for $x < \frac{1}{2}$

We now treat the first case of the recursion in (A.2). We do this by establishing a bijection between the solutions to (A.1). Let us denote solutions belonging to $\frac{\alpha}{\gamma}$ by a subscript 1 and those belonging to $\frac{\alpha}{\gamma - \alpha}$ by a subscript 2. Assume that we have a solution to $\frac{\alpha}{\gamma}$, i.e.

$$\begin{pmatrix} \alpha \\ \gamma \end{pmatrix} = \begin{pmatrix} \tilde{d}_1 & \tilde{b}_1 \\ \tilde{c}_1 & \tilde{a}_1 \end{pmatrix} \begin{pmatrix} a \\ c \end{pmatrix}. \qquad \text{(A.5)}$$

We then set

$$\begin{pmatrix} \tilde{d}_2 & \tilde{b}_2 \\ \tilde{c}_2 & \tilde{a}_2 \end{pmatrix} = \begin{pmatrix} 1 & 0 \\ -1 & 1 \end{pmatrix} \begin{pmatrix} \tilde{d}_1 & \tilde{b}_1 \\ \tilde{c}_1 & \tilde{a}_1 \end{pmatrix}, \tag{A.6}$$

which by construction satisfies

$$\begin{pmatrix} \alpha \\ \gamma - \alpha \end{pmatrix} = \begin{pmatrix} \tilde{d}_2 & \tilde{b}_2 \\ \tilde{c}_2 & \tilde{a}_2 \end{pmatrix} \begin{pmatrix} a \\ c \end{pmatrix}. \tag{A.7}$$

To check that this defines a bijection, we have to show that the bound $g^{-1} \cdot \infty < -1$ is mapped correctly, which as we said above amount to say that $\frac{\tilde{a}}{\tilde{c}} > 1$. Thus we need to check that the two bounds imply each other,

$$\frac{\tilde{a}_1}{\tilde{c}_1} > 1 \iff \frac{\tilde{a}_2}{\tilde{c}_2} > 1. \tag{A.8}$$

$\Longrightarrow$: We first go from left to right and check that $\tilde{a}_2 > \tilde{c}_2$ and $\tilde{c}_2 > 0$. This follows from

$$\tilde{a}_2 - \tilde{c}_2 = \tilde{a}_1 - \tilde{c}_1 - \tilde{b}_1 + \tilde{d}_1 = \frac{a + c + (\tilde{a}_1 - \tilde{c}_1)(\gamma - \alpha)}{\gamma} > 0, \tag{A.9}$$

where we used (A.5) and $\tilde{a}_1 - \tilde{c}_1 > 0$ and $\gamma - \alpha > 0$. Bounding $\tilde{c}_2$ is a bit more tricky and we compute

$$\tilde{c}_2 = \tilde{c}_1 - \tilde{d}_1 = -\frac{\gamma - a\tilde{c}_1}{\tilde{a}_1 \gamma} + \tilde{c}_1 \frac{\gamma - \alpha}{\gamma} \geqslant -\frac{\gamma - \tilde{c}_1}{\tilde{a}_1 \gamma} + \tilde{c}_1 \frac{\gamma - \alpha}{\gamma}. \tag{A.10}$$

Equality of the first and second line follows from (A.5). We also used that the expression becomes biggest for the smallest possible value of $a$ which is $a = 1$. We now also minimize this expression over $\tilde{a}_1$ and $\tilde{c}_1$. It is minimized for the smallest possible value of $\tilde{a}_1$, which is $\tilde{c}_1 + 1$. Thus

$$\tilde{c}_2 \geqslant -\frac{\gamma - \tilde{c}_1}{(\tilde{c}_1 + 1)\gamma} + \tilde{c}_1 \frac{\gamma - \alpha}{\gamma}. \tag{A.11}$$

This expression becomes positive for large values of $\tilde{c}_1$. By computing the derivative, we see that it doesn't have an extremum on the real line. Thus it becomes smallest for the smallest allowable value of $\tilde{c}_1$, which is $\tilde{c}_1 = 1$, where it becomes

$$\tilde{c}_2 \geqslant \frac{1 + \gamma - 2\alpha}{2\gamma} > 0, \tag{A.12}$$

where we finally used our assumption that $2\alpha < \gamma$.

$\Longleftarrow$: We also need to check that this also works the other way round. Assume that $\tilde{c}_2 > 0$ and $\tilde{a}_2 - \tilde{c}_2 > 0$. We then need to check that $\tilde{c}_1 > 0$ and $\tilde{a}_1 - \tilde{c}_1 > 0$. This is doable with the same tricks as above:

$$\tilde{c}_1 = \tilde{c}_2 + \tilde{d}_2 = \frac{\tilde{c}_2 \gamma + c}{\gamma - \alpha} > 0, \tag{A.13}$$

$$\tilde{a}_1 - \tilde{c}_1 = -\frac{1}{\tilde{c}_2} + \frac{(\tilde{a}_2 - \tilde{c}_2)(c + \tilde{c}_2 \gamma)}{\tilde{c}_2(\gamma - \alpha)} \geqslant -\frac{1}{\tilde{c}_2} + \frac{1 + \tilde{c}_2 \gamma}{\tilde{c}_2(\gamma - \alpha)} \geqslant \frac{\alpha + 1}{\gamma - \alpha} > 0, \tag{A.14}$$

where we used that the expression is minimized for $\tilde{a}_2 = \tilde{c}_2 + 1$ and $c = 1$. The remaining expression is then minimized at $\tilde{c}_2 = 1$.

Thus we see that solutions belonging to $\frac{\alpha}{\gamma}$ and solutions belonging to $\frac{\alpha}{\gamma - \alpha}$ can be mapped to each other bijectively, provided that $\frac{\alpha}{\gamma} < \frac{1}{2}$. This demonstrates the first case of the recursion relation (A.2).

## A.3   Recursion for $x > \frac{1}{2}$

Now consider the case $\frac{\alpha}{\gamma} > \frac{1}{2}$. We again want to map solutions to solutions for $\frac{2\alpha - \gamma}{\alpha}$. This should work, but for one exception. We set

$$\begin{pmatrix} \tilde{d}_2 & \tilde{b}_2 \\ \tilde{c}_2 & \tilde{a}_2 \end{pmatrix} = \begin{pmatrix} 2 & -1 \\ 1 & 0 \end{pmatrix} \begin{pmatrix} \tilde{d}_1 & \tilde{b}_1 \\ \tilde{c}_1 & \tilde{a}_1 \end{pmatrix}, \tag{A.15}$$

which clearly maps the condition $x = g \cdot y$ correctly. We thus need to look at the condition $g^{-1} \cdot \infty < -1$ and check that it maps to the correct range. This should work, except for one exception which leads to the $+1$ in the recursion relation (A.2).

$\Longrightarrow$:   Assume that $\tilde{a}_1 > \tilde{c}_1 > 0$. We need to check the inequalities $\tilde{a}_2 > \tilde{c}_2 > 0$. This is trivial for $\tilde{c}_2$:

$$\tilde{c}_2 = \tilde{d}_1 = \frac{\alpha \tilde{c}_1 + c}{\gamma} > 0. \tag{A.16}$$

For the other inequality, assume first that $\frac{\tilde{a}_1}{\tilde{c}_1} \neq 2$, i.e. either $\tilde{c}_1 \geqslant 2$ or $\tilde{a}_1 - \tilde{c}_1 \geqslant 2$ holds. Then

$$\tilde{a}_2 - \tilde{c}_2 = \tilde{b}_1 - \tilde{d}_1 \tag{A.17a}$$

$$= \frac{(\tilde{a}_1 - \tilde{c}_1)(c + \tilde{c}_1 \alpha)}{\tilde{c}_1 \gamma} - \frac{1}{\tilde{c}_1} \tag{A.17b}$$

$$\geqslant \frac{(\tilde{a}_1 - \tilde{c}_1)(1 + \tilde{c}_1 \alpha)}{\tilde{c}_1 \gamma} - \frac{1}{\tilde{c}_1} \tag{A.17c}$$

$$\geqslant \min\left( \frac{2(1 + \tilde{c}_1 \alpha)}{\tilde{c}_1 \gamma} - \frac{1}{\tilde{c}_1}\bigg|_{\tilde{c}_1 \geqslant 1}, \frac{1 + \tilde{c}_1 \alpha}{\tilde{c}_1 \gamma} - \frac{1}{\tilde{c}_1}\bigg|_{\tilde{c}_1 \geqslant 2} \right) \tag{A.17d}$$

$$\geqslant \min\left( \frac{2\alpha - \gamma + 2}{\gamma}, \frac{2\alpha - \gamma + 1}{2\gamma} \right) > 0. \tag{A.17e}$$

For $\tilde{a}_1 = 2$ and $\tilde{c}_1 = 1$ this argument fails. In this case, there is a unique solution for $a$ and $c$. Indeed, the conditions relating $\frac{a}{c}$ and $\frac{\alpha}{\gamma}$ can be solved to give

$$a = 2\alpha - (2\tilde{d}_1 - 1)\gamma, \qquad c = -\alpha + \tilde{d}_1 \gamma. \tag{A.18}$$

Imposing $a > 0$ and $c > 0$ leaves $\tilde{d}_1 = 1$ as the unique solution and so

$$a = 2\alpha - \gamma, \qquad c = \gamma - \alpha. \tag{A.19}$$

This solution would map to $\tilde{a}_2 = \tilde{c}_2 = 1$ under (A.15), which is forbidden. Thus we conclude that all solutions, but this one, map to solutions belonging to $\frac{2\alpha - \gamma}{\alpha}$.

$\Longleftarrow$:   We also need to check the inverse direction. So suppose that $\tilde{a}_2 > \tilde{c}_2 > 0$. We aim to show that $\tilde{a}_1 > \tilde{c}_1 > 0$. We follow the same strategy as above,

$$\tilde{c}_1 = 2\tilde{c}_2 - \tilde{d}_2 = \frac{\tilde{c}_2(\tilde{a}_2 \gamma + a) - \alpha}{\tilde{a}_2 \alpha} \geqslant \frac{(\tilde{a}_2 - 1)(\tilde{a}_2 \gamma + 1) - \alpha}{\tilde{a}_2 \alpha} \geqslant \frac{2\gamma - \alpha + 1}{2\alpha} > 0, \tag{A.20}$$

$$\tilde{a}_1 - \tilde{c}_1 = (2\tilde{a}_2 - \tilde{b}_2) - (2\tilde{c}_2 - \tilde{d}_2) = \frac{(\tilde{a}_2 - \tilde{c}_2)\gamma + a + c}{\alpha} > 0. \tag{A.21}$$

Thus all solutions of $\frac{\alpha}{\gamma}$ except for the special one with $\frac{a}{c} = \frac{2\alpha - \gamma}{\gamma - \alpha}$ and $\frac{\tilde{a}}{\tilde{c}} = 2$ are in bijection with the solutions of $\frac{2\alpha - \gamma}{\alpha}$. This demonstrates the second case of the recursion relation (A.2).

# B Convergence

In this appendix, we demonstrate that the Rademacher expansion developed in this paper converges, provided that there exists $\varepsilon > 0$ such that

$$|\tau + \tilde\tau|^{2+\varepsilon} |\eta(\tau)\eta(\tilde\tau)|^{2\varepsilon} |f(\tau,\tilde\tau)| \leqslant C, \qquad \tau \in C_{a/c},\ \tilde\tau \in C_{\tilde a/\tilde c}, \tag{B.1}$$

for some uniform constant $C$ independent of $a$, $c$, $\tilde a$, $\tilde c$ and the locations of $\tau$, $\tilde\tau$ on the Ford circles. Here we using the modular covariant function as in (5). This bound is independent of the modular frame. In other words, the integrand is uniformly bounded on the Rademacher circles and slowly decays as we approach the cusp, due to the presence of the eta-functions in (B.1).

## B.1 Dominated convergence

The idea for the following proof is to use the dominated convergence theorem. The contour deformations that bring us to the Rademacher contour depend on the integer $n$ as described in sections 2.4 and 2.5. The contours get successively longer, i.e. $\Gamma_{n+1} = \Gamma_n \cup \delta_n$, where $\delta_n$ is a contractible contour. Instead of deforming the contour, we can directly integrate over the limiting contour $\Gamma_\infty$, but include the characteristic function $\mathbb{1}_n$ in the integrand. It is by definition 1 on the contour $\Gamma_n$ and 0 on $\Gamma_\infty \setminus \Gamma_n$. Thus we can write e.g. the first contribution obtained in (39) as

$$\frac{1}{6}(\tau - \tilde\tau) \quad = \frac{1}{6}\lim_{n\to\infty} \int_{\tau\in\longrightarrow} \int_{\tilde\tau\in\Gamma_\infty} \mathbb{1}_n(\tilde\tau)(\tau-\tilde\tau)\alpha(\tau,\tilde\tau). \tag{B.2}$$

The equality is true even without taking the limit $n \to \infty$, as long as $n \geqslant 1$, but we may in particular take the limit. Thus, convergence boils down to commuting the integral with the limit. For this purpose, we can use the dominated convergence theorem, which tells us that this exchange is legal if the absolute value of the integrand is bounded by a function independent of $n$, which is integrable. Thus the Rademacher procedure converges to the correct integral provided that

$$\int_{\tau\in\longrightarrow} \int_{\tilde\tau\in\Gamma_\infty} |(\tau-\tilde\tau)\alpha(\tau,\tilde\tau)| < \infty, \tag{B.3}$$

and similarly for the other pieces of the contour. Thus a sufficient criterion for convergence is the convergence of (2) with absolute values around the integrand,

$$\sum_{c=1}^{\infty} \sum_{\substack{a=0 \\ (a,c)=1}}^{c-1} \int_{\tau\in\longrightarrow} \int_{\tilde\tau\in C_{a/c}} \left| \left[ \frac{1}{6}\left(\tau - \tilde\tau + \frac{2a}{c}\right) - 2s(a,c) \right] \alpha(\tau,\tilde\tau) \right| < \infty. \tag{B.4}$$

The strategy will be to first bound the integrals as a power of $c$, which in turn will prove convergence of the infinite sum.

## B.2 Bounding integrals over Ford circles

By using the triangle inequality, we can check (B.4) term by term. Assume that $\varepsilon > 0$ is fixed. We first bound the integral

$$I_{a/c}^{(1)} = \int_{\tau\in\longrightarrow} \int_{\tilde\tau\in C_{a/c}} |\alpha(\tau,\tilde\tau)|. \tag{B.5}$$

We write $\lesssim$ for a bound in absolute value up to a constant independent of $a$ and $c$. We have with the help of (B.1) and the fact that $|\eta(\tau)|$ is bounded on the horizontal contour:

$$\left|I_{a/c}^{(1)}\right| \lesssim \int_{\tau\in\longrightarrow} \int_{\tilde{\tau}\in C_{a/c}} \frac{|\mathrm{d}\tau\,\mathrm{d}\tilde{\tau}|}{|\tau+\tilde{\tau}|^{2+\varepsilon}\,|\eta(\tilde{\tau})|^{2\varepsilon}} \tag{B.6a}$$

$$= \int_{\tau\in\longrightarrow} \int_{\tilde{\tau}\in\longrightarrow} \frac{|\mathrm{d}\tau\,\mathrm{d}\tilde{\tau}|}{|c\tilde{\tau}+d|^{2+\varepsilon}\left|\tau+\frac{a\tilde{\tau}+b}{c\tilde{\tau}+d}\right|^{2+\varepsilon}|\eta(\tilde{\tau})|^{2\varepsilon}} \tag{B.6b}$$

$$= \int_{\tau\in\longrightarrow} \int_{\tilde{\tau}\in\longrightarrow} \frac{|\mathrm{d}\tau\,\mathrm{d}\tilde{\tau}|}{|c\tilde{\tau}|^{2+\varepsilon}\left|\tau-\frac{1}{c\tilde{\tau}}\right|^{2+\varepsilon}|\eta(\tilde{\tau})|^{2\varepsilon}} \tag{B.6c}$$

$$\lesssim \int_{\tau\in\longrightarrow} \int_{\tilde{\tau}\in\longrightarrow} \frac{|\mathrm{d}\tau\,\mathrm{d}\tilde{\tau}|}{\left|c\tau\tilde{\tau}-\frac{1}{c}\right|^{2+\varepsilon}} \tag{B.6d}$$

$$\lesssim c^{-2-\varepsilon} \int_{\tau\in\longrightarrow} \int_{\tilde{\tau}\in\longrightarrow} \frac{|\mathrm{d}\tau\,\mathrm{d}\tilde{\tau}|}{\left(|\tau\tilde{\tau}|-\frac{1}{c_0^2}\right)^{2+\varepsilon}} \tag{B.6e}$$

$$\lesssim c^{-2-\varepsilon}\,. \tag{B.6f}$$

We followed the same kind of steps as in the evaluation of the Rademacher formula in the examples discussed in section 3. In the penultimate step, we used the inverse triangle inequality and assumed that $c \geqslant c_0 > 1$ as the cases with $c < c_0$ may be dealt with by making the overall constant bigger. We also need to bound the integral

$$I_{a/c}^{(2)} = \int_{\tau\in\longrightarrow} \int_{\tilde{\tau}\in C_{a/c}} \tau\alpha(\tau,\tilde{\tau})\,. \tag{B.7}$$

Following the same strategy, we have

$$|I_{a/c}^{(2)}| \lesssim c^{-2-\varepsilon} \int_{\tau\in\longrightarrow} \int_{\tilde{\tau}\in\longrightarrow} \frac{|\tau||\mathrm{d}\tau\,\mathrm{d}\tilde{\tau}|}{\left(|\tau\tilde{\tau}|-\frac{1}{c_0^2}\right)^{2+\varepsilon}} \lesssim c^{-2-\varepsilon}\,. \tag{B.8}$$

Since $\varepsilon > 0$, it follows immediately that

$$\sum_{c=1}^{\infty} \sum_{\substack{a=0\\(a,c)=1}}^{c-1} \left|I_{a/c}^{(j)}\right| \lesssim \sum_{c=1}^{\infty} \frac{1}{c^{1+\varepsilon}} < \infty\,, \qquad j = 1, 2\,. \tag{B.9}$$

Since $\tilde{\tau}$ and $\frac{a}{c}$ are both bounded on the integration contour, this takes care of the absolute convergence of the contribution of $\frac{1}{6}(\tau - \tilde{\tau} + \frac{2a}{c})$ in (B.4).

It remains to bound the last term in (B.4) involving $s(a,c)$. For this, we can use the known bound [47]

$$\sum_{\substack{a=0\\(a,c)=1}}^{c-1} |s(a,c)| \leqslant \frac{1}{2\pi^2}\varphi(c)\log^2(c) < \frac{c}{2\pi^2}\log^2(c)\,, \tag{B.10}$$

valid for sufficiently large $c$. Here $\varphi(c)$ is the Euler totient function. It then follows that the last term in (B.4) is also absolutely convergent:

$$\sum_{c=1}^{\infty} \sum_{\substack{a=0\\(a,c)=1}}^{c-1} \left|s(a,c)\,I_{a/c}^{(1)}\right| \lesssim \sum_{c=1}^{\infty} \frac{\log^2(c)}{c^{1+\varepsilon}} < \infty\,. \tag{B.11}$$

## C  Rademacher expansion using a different Lorentzian integration formula

In this appendix, we derive the alternative Lorentzian integration formulae (66a), (66b) and (66c). To have some variety, we will work with $\tau$ and $\overline{\tau}$ and with a scalar integrand $\frac{\mathrm{d}^2\tau}{(\mathrm{Im}\,\tau)^2}\hat{f}(\tau,\overline{\tau})$. We will also denote $\tau_2 = \mathrm{Im}\,\tau$.

### C.1  Derivation of the Lorentzian integration formula

We express the Hauptmodul $z = \lambda(\tau) = \frac{\vartheta_2(\tau)^4}{\vartheta_3(\tau)^4}$ for $\Gamma(2)$ in terms of the $\rho$ variable defined by $z = \frac{4\rho}{(1+\rho)^2}$ so that[10]

$$\tau = i\,\frac{K\left(1 - \frac{4\rho}{(1+\rho)^2}\right)}{K\left(\frac{4\rho}{(1+\rho)^2}\right)}, \qquad \overline{\tau} = -i\,\frac{K\left(1 - \frac{4\overline{\rho}}{(1+\overline{\rho})^2}\right)}{K\left(\frac{4\overline{\rho}}{(1+\overline{\rho})^2}\right)}. \tag{C.2}$$

The $\rho$ variable lives on the double cover of the complex $z$-plane branched across $z \in [1,\infty)$. The relation (C.2) is a one-to-one map from the interior of the unit $\rho$ disk to the interior of the fundamental domain for $\Gamma(2)$ denoted $\mathcal{F}_2$, which we discussed in section 2.2. Next, we define express $\rho$ in terms of radial and angular coordinates,

$$\rho = \sigma w^{-1}, \qquad \overline{\rho} = \sigma w, \tag{C.3}$$

with $\sigma \in (0,1)$ and $|w| = 1$. We can therefore express the integral of a modular invariant function $\hat{f}$ over $\mathcal{F}_2$ as

$$\int_{\mathcal{F}_2} \frac{\mathrm{d}^2\tau}{\tau_2^2}\hat{f}(\tau,\overline{\tau}) = \int_0^1 \mathrm{d}\sigma \oint \mathrm{d}w\, J(\sigma,w)\frac{\hat{f}(\tau(\sigma,w),\overline{\tau}(\sigma,w))}{\tau_2^2(\sigma,w)}, \tag{C.4}$$

where the Jacobian $J$ is given by

$$J(\sigma,w) = \frac{\sigma}{iw}\frac{\partial\tau}{\partial\rho}\frac{\partial\overline{\tau}}{\partial\overline{\rho}}. \tag{C.5}$$

We now identify the singularities of the integrand in the complex $w$-plane. There are branch points at

$$w = 0,\ \pm\sigma,\ \pm\frac{1}{\sigma},\ \infty, \tag{C.6}$$

arising from the Hauptmodul alone. For the moment, we shall be agnostic about the singularities of $\hat{f}$ at the cusps as they can be easily regulated using the $i\varepsilon$-prescription which we shall incorporate at the end of the calculation. In the complex $z$-plane, using $\rho = \frac{z}{(1-\sqrt{1-z})^2}$ for $z$ on the second sheet reached by a monodromy around the branch point at $z = 1$, we see that the $w \in (-\sigma,\sigma)$ cut corresponds to image of the interval $z \in (-\infty,1]$ in the second sheet(s). Similarly, using $\overline{\rho} = \frac{\overline{z}}{(1+\sqrt{1-\overline{z}})^2}$ for $\overline{z}$ in the first sheet, we see that the $w \in (-\sigma,\sigma)$ cut corresponds to image of the interval $\overline{z} \in (-\infty,1]$ in the first sheet. For the other cuts, $w \in (\frac{1}{\sigma},\infty)$ and $w \in (-\infty,-\frac{1}{\sigma})$, the mappings are similar with $z \longleftrightarrow \overline{z}$. Now, we can deform

---

[10]Note that $K(z)$ is given by an elliptic integral which evaluates to the hypergeometric function,

$$K(z) = \frac{\pi}{2}\,_2F_1\left(\tfrac{1}{2},\tfrac{1}{2};1;z\right). \tag{C.1}$$

the $w$ contour in (C.4) from the unit circle to wrap the branch cut, $w \in (-\sigma, \sigma)$. We thus have from (C.4),

$$I := \int_{\mathcal{F}} \frac{d^2 \tau}{\tau_2^2} \hat{f}(\tau, \overline{\tau}) = \frac{1}{6} \int_0^1 d\sigma \int_{-\sigma}^{\sigma} dw \, \mathrm{Disc}_w \left[ J(\sigma, w) \frac{\hat{f}(\tau(\sigma, w), \overline{\tau}(\sigma, w))}{\tau_2^2(\sigma, w)} \right]. \qquad (C.7)$$

Along the interval $(0, \sigma)$, $\rho \in (1, \infty)$ and $\overline{\rho} \in (0, 1)$ (as $\sigma$ is varied from 0 to 1 and subject to $\rho \overline{\rho} \leqslant 1$). Since $z \circlearrowleft 1$ or $z \circlearrowright 1$ corresponds to $\rho \to \frac{1}{\rho}$, this means $z \in (0, 1)$ in the second sheet and $\overline{z} \in (0, 1)$ in the first sheet. (The $\rho \overline{\rho} \leqslant 1$ constraint translates to $\overline{z} \in (0, z)$. However, the $z \longleftrightarrow \overline{z}$ symmetry in the problem allows us extend the domain to $\overline{z} \in (0, 1)$ with a factor of $\frac{1}{2}$. Note that this constraint would be trivial if both variables are on the first sheet) Observe that under (C.2), $\rho \in (0, 1)$ maps to the imaginary $\tau$ axis downward. Going to the second sheet(s) sends $\tau \to \frac{\tau}{\pm 2\tau+1}$. Similarly, along the interval $w \in (-\sigma, 0)$, $\rho \in (-\infty, -1)$ and $\overline{\rho} \in (-1, 0)$. Observe that under (C.2), $\rho \in (-1, 0)$ maps to the vertical boundaries of the $\mathcal{F}_2$. This corresponds to $z \in (-\infty, 0)$ on the second sheet and $\overline{z} \in (-\infty, 0)$ on the first sheet. (Note that $(-\infty, 0)$ is the image of $(0, 1)$ under the $s$-$u$ crossing transformation $z \to \frac{z}{z-1}$). Thus, from (C.7) we have,

$$6I = 2 \int_0^{i\infty} d\tau \int_{-i\infty}^0 d\overline{\tau} \frac{1}{2i} \left[ \frac{\hat{f}\left(\frac{\tau}{-2\tau+1}, \overline{\tau}\right)}{\left(\frac{\tau}{-2\tau+1} - \overline{\tau}\right)^2 (-2\tau+1)^2} - \frac{\hat{f}\left(\frac{\tau}{2\tau+1}, \overline{\tau}\right)}{\left(\frac{\tau}{2\tau+1} - \overline{\tau}\right)^2 (2\tau+1)^2} \right]$$

$$+ 2 \int_0^{i\infty} d\tau \int_{-i\infty}^0 d\overline{\tau} \frac{1}{2i} \left[ -\frac{\hat{f}\left(\frac{\tau+1}{-2\tau-1}, \overline{\tau}-1\right)}{\left(\frac{\tau+1}{-2\tau-1} - (\overline{\tau}-1)\right)^2 (2\tau+1)^2} \right. \qquad (C.8)$$

$$\left. + \frac{\hat{f}\left(\frac{\tau-1}{2\tau-1}, \overline{\tau}+1\right)}{\left(\frac{\tau-1}{2\tau-1} - (\overline{\tau}+1)\right)^2 (-2\tau+1)^2} \right].$$

A $T$-transformation of the arguments of $\hat{f}$ in the second line makes the second line equal to the first line. This $s$-$u$ crossing symmetry can be explained by the following features of the elliptic map. $z = \lambda(\tau) \implies \rho^2 = \lambda(2\tau)$ with $\rho$ on the first sheet and $2\tau \in \mathcal{F}_2$. Inverting this expression gives $\tau = n + \frac{i}{2} \frac{K(1-\rho^2)}{K(\rho^2)}$ with $n \in \mathbb{Z}$. The ambiguity is because of the periodicity, $\lambda(2\tau) = \lambda(2\tau \pm 2)$. It is fixed by requiring that $2\tau \in \mathcal{F}_2$. Since $s$-$u$ crossing corresponds to $\rho \to -\rho$, which by the above features corresponds at most to a modular transformation on $\tau$, $s$-$u$ crossing is indeed a symmetry on the first sheet. However, to explain our result, we require it to also be a symmetry on the second sheet(s). This can be explained by another feature of the map, $\lambda(\frac{\tau}{\pm\tau+1}) = \frac{1}{\lambda(\tau)}$. This shows that under the monodromy transformation, $\tau \to \frac{\tau}{\pm 2\tau+1}$, $\rho^2 \to \frac{1}{\rho^2}$. An $S$-transformation, $\hat{f}\left(\frac{\tau}{\pm 2\tau+1}, \overline{\tau}\right) = \hat{f}\left(-\frac{1}{\tau} \mp 2, -\frac{1}{\overline{\tau}}\right)$ followed by a change of variables $\tau \to -\frac{1}{\tau}$ and $\overline{\tau} \to -\frac{1}{\overline{\tau}}$ brings the above integral to the form,

$$\int_{\mathcal{F}} \frac{d^2 \tau}{\tau_2^2} \hat{f}(\tau, \overline{\tau})$$

$$= \frac{2}{3} \int_0^{i\infty} d\tau \int_{-i\infty}^0 d\overline{\tau} \frac{1}{2i} \left[ (\tau+2-\overline{\tau})^{-2} \hat{f}(\tau+2, \overline{\tau}) - (\tau-2-\overline{\tau})^{-2} \hat{f}(\tau-2, \overline{\tau}) \right], \qquad (C.9)$$

which is the Lorentzian integral formula that we sought to derive. Translating to our previous conventions gives (66a).

There is an alternate representation of this formula which can be derived by a contour deformation. We deform the integration contour in $\tau$ between $\pm 2$ to the semicircles of unit

radius on either side of the imaginary axis. This gives

$$
6I = 4 \int_0^{i\infty} d\tau \int_{-i\infty}^0 d\overline{\tau} \, \frac{1}{2i} \left[ \frac{\hat{f}\left(\frac{2\tau}{-\tau+2}, \overline{\tau}\right)}{\left(\frac{2\tau}{-\tau+2} - \overline{\tau}\right)^2 (\tau - 2)^2} - \frac{\hat{f}\left(\frac{2\tau}{\tau+2}, \overline{\tau}\right)}{\left(\frac{2\tau}{\tau+2} - \overline{\tau}\right)^2 (\tau + 2)^2} \right] . \tag{C.10}
$$

An $S$-transformation followed by a change of variables $\tau \to -\frac{1}{\tau}$ and $\overline{\tau} \to -\frac{1}{\overline{\tau}}$ brings the above integral to the form,

$$
\begin{aligned}
&\int_{\mathcal{F}} \frac{d^2\tau}{\tau_2^2} \hat{f}(\tau, \overline{\tau}) \\
&= \frac{2}{3} \int_0^{i\infty} d\tau \int_{-i\infty}^0 d\overline{\tau} \, \frac{1}{2i} \left[ \left(\tau + \tfrac{1}{2} - \overline{\tau}\right)^{-2} \hat{f}\left(\tau + \tfrac{1}{2}, \overline{\tau}\right) - \left(\tau - \tfrac{1}{2} - \overline{\tau}\right)^{-2} \hat{f}\left(\tau - \tfrac{1}{2}, \overline{\tau}\right) \right],
\end{aligned} \tag{C.11}
$$

which is equivalent to (66b).

## C.2 Lorentzian integration using the $J$-function

We will now derive the third integration formula (66c). It is known that the Klein-invariant $J(\tau)$ serves as the Hauptmodul between the fundamental domain for the modular group $\Gamma$ and the complex plane. Just like in the $\Gamma(2)$ case, the inverse is expressed as a ratio of hypergeometric functions,

$$
J(\tau) = \frac{1}{4z(1-z)} \implies \tau = i \frac{{}_2F_1\left(\frac{1}{6}, \frac{5}{6}; 1; 1-z\right)}{{}_2F_1\left(\frac{1}{6}, \frac{5}{6}; 1; z\right)} . \tag{C.12}
$$

The above expression for $\tau$ corresponds to a $1-1$ map between $\mathcal{F} \cup S\mathcal{F}$ and the complex $z$-plane. Under this map, the monodromy around $z = 1$ corresponds to $\tau \to \frac{\tau}{\pm\tau+1}$. The intervals in $z$, $(0, 1)$, $(1, \infty)$, $(-\infty, 0)$ map respectively to the imaginary $\tau$ axis, the circular arcs along the boundary of $\mathcal{F} \cup S\mathcal{F}$ and the vertical boundaries of $\mathcal{F} \cup S\mathcal{F}$. By following similar steps as in the $\Gamma(2)$ case, we arrive at the following Lorentzian formula for the integral of a modular invariant function over the fundamental domain,

$$
\begin{aligned}
&\int_{\mathcal{F}} \frac{d^2\tau}{\tau_2^2} \hat{f}(\tau, \overline{\tau}) \\
&= 2 \int_0^{i\infty} d\tau \int_{-i\infty}^0 d\overline{\tau} \, \frac{1}{2i} \left[ \left(\tau - \omega^2 - \overline{\tau}\right)^{-2} \hat{f}\left(\tau - \omega^2, \overline{\tau}\right) - \left(\tau + \omega - \overline{\tau}\right)^{-2} \hat{f}\left(\tau + \omega, \overline{\tau}\right) \right],
\end{aligned} \tag{C.13}
$$

where $\omega = e^{\frac{2\pi i}{3}}$ is a cube root of unity. Notice that $\omega$ and $-\omega^2$ are the two corners in the fundamental domain $\mathcal{F}$. Again, the above result should be made sense of by incorporating the $i\varepsilon$-prescription just like in the previous cases. In the notation of the main text,

$$
2I \overset{\text{Re}}{=} \left( \int_{\tau \in \Gamma} \int_{\tilde{\tau} \in \omega + \Gamma} - \int_{\tau \in \Gamma} \int_{\tilde{\tau} \in -\omega^2 + \Gamma} \right) \alpha(\tau, \tilde{\tau}), \tag{C.14}
$$

where $\Gamma$ is the contour also appearing in (24).

## C.3 The Rademacher expansion

We will now indicate the modifications necessary for the derivation of the Rademacher formula starting from the Lorentzian integration formula (66a). We start with the Lorentzian formula written in terms of the differential form $\alpha$ and incorporating the $i\varepsilon$-prescription to regulate the divergences at the cusps,

$$
6I \overset{\text{Re}}{=} \left( \int_{\tau \in \Gamma} \int_{\tilde{\tau} \in -2 + \Gamma} - \int_{\tau \in \Gamma} \int_{\tilde{\tau} \in 2 + \Gamma} \right) \alpha(\tau, \tilde{\tau}), \tag{C.15}
$$

where $I$ denotes the principal value of the integral over the fundamental domain i.e., the average of the two $i\varepsilon$-prescriptions. The contour $\Gamma$ is the same as the $\tau$-contour in figure 4. After a series of contour deformations, we can express the final answer in terms of a sum over Ford circles as

$$I = \frac{1}{6} \sum_{c=1}^{\infty} \sum_{\substack{a=0 \\ (a,c)=1}}^{c-1} \int_{\tau \in \longrightarrow} \int_{\tilde{\tau} \in C_{a/c}} \left( \tau - \tilde{\tau} + \tilde{\mu}\left(\frac{a}{c}\right) \right) \alpha(\tau, \tilde{\tau}), \tag{C.16}$$

where $\tilde{\mu}$ is given by

$$\tilde{\mu}(x) = \begin{cases} 0, & x = 0, \\ \tilde{m}(x) - \tilde{m}(1-x), & 0 < x < \frac{1}{2}, \\ 1, & x = \frac{1}{2}, \\ \tilde{m}(x) - \tilde{m}(1-x) + 2, & \frac{1}{2} < x < 1, \end{cases} \tag{C.17}$$

and $\tilde{m}$ is defined in (C.24). We observed numerically that when the values of $\tilde{\mu}$ are averaged over their values at $\frac{a}{c}$ and $\frac{a^*}{c}$, it agrees with the Rademacher function,

$$\frac{1}{2}\left( \tilde{\mu}\left(\frac{a}{c}\right) + \tilde{\mu}\left(\frac{a^*}{c}\right) \right) = \Phi\begin{pmatrix} a & b \\ c & a^* \end{pmatrix} = \mu\left(\frac{a}{c}\right), \tag{C.18}$$

so (C.16) matches with (63a) thanks to (62c). We have verified this numerically up to $c = 200$. It is presumably not hard to prove this along the same lines as we did for $\mu$, but we haven't tried to do so since it is evident that we will obtain the same Rademacher formula.

To derive the formula (C.16), we follow steps very similar to the ones outlined in section 2 with minor modifications. So we shall be very brief.

### C.3.1   First term of (C.15)

$$\overset{\text{Re}}{=} \tag{C.19}$$

After further manipulations of the first term on the RHS above, we get

$$\overset{\text{Re}}{=} (\tau - \tilde{\tau}) \tag{C.20}$$

### C.3.2   Second term of (C.15)

$$\overset{\text{Re}}{=} - \quad + 2 \tag{C.21}$$

### C.3.3   Rademacherization

Combining the two terms, we have

$$I \overset{\text{Re}}{=} \frac{1}{6}(\tau - \tilde{\tau}) \quad \text{[contour]} \quad - \frac{1}{3} \quad \text{[contour]} \quad + \frac{1}{3} \quad \text{[contour]} \quad . \tag{C.22}$$

Now, we can Rademacherize the contours to get

$$
\begin{aligned}
I \overset{\text{Re}}{=} & \frac{1}{6} \sum_{c=1}^{\infty} \sum_{\substack{a=0 \\ (a,c)=1}}^{c-1} \int_{\tau \in \longrightarrow} \int_{\tilde{\tau} \in C_{a/c}} \left( \tau - \tilde{\tau} + 2\tilde{m}\left(\frac{a}{c}\right) \right) \alpha(\tau, \tilde{\tau}) \\
& + \frac{1}{3} \sum_{c=1}^{\infty} \sum_{\substack{(a,c)=1}}^{\frac{1}{2} < \frac{a}{c} < 1} \int_{\tau \in \longrightarrow} \int_{\tilde{\tau} \in C_{a/c}} \alpha(\tau, \tilde{\tau}),
\end{aligned}
\tag{C.23}
$$

where $\tilde{m}$ is defined as

$$\tilde{m}\left(\frac{\alpha}{\gamma}\right) = \# \left\{ \frac{a}{c} \in \mathbb{Q}_{>0}, \, \frac{\tilde{a}}{\tilde{c}} \in \mathbb{Q}_{>2} \, \bigg| \, \frac{\alpha}{\gamma} \equiv \frac{a\tilde{d} + \tilde{b}c}{\tilde{a}c + a\tilde{c}} \mod 1 \right\}. \tag{C.24}$$

Counting the Ford circles numerically, we get the table 2 for values of $\tilde{m}$. Experimentally, the following recursion relation seems to hold

$$
\tilde{m}(x) = \begin{cases}
\tilde{m}(\frac{x}{1-x}), & x < \frac{1}{2}, \\
\tilde{m}(\frac{2x-1}{x}), & \frac{1}{2} \le x \le \frac{2}{3}, \\
\tilde{m}(\frac{2x-1}{x}) + 1, & x > \frac{2}{3},
\end{cases}
\tag{C.25}
$$

with initial condition $\tilde{m}(0) = 0$. This is very similar to the recursion relation for $m$ (44). It presumably can be proven along the same lines of appendix A, but we have not tried to do so. We checked this numerically up to $c = 200$. Evaluating the real part of the RHS in (C.23), we get (C.16).

# D Two integrals

We will now derive the two integral formulas (75) and (76) that appear in all the applications.

## D.1 First integral

We will start with $\mathcal{I}_c^{(1)}(n, x, \tilde{x})$,

$$\mathcal{I}_c^{(1)}(n, x, \tilde{x}) = \int_{\longrightarrow} d\tau \int_{\longrightarrow} d\tilde{\tau} \frac{e^{-ix\tau - i\tilde{x}\tilde{\tau}}}{\left(\frac{1}{c} - c\tau\tilde{\tau}\right)^n}. \tag{D.1}$$

The first step is to make a change of variables to $\tau = y + \frac{1}{c^2\tilde{\tau}}$ to get

$$\mathcal{I}_c^{(1)}(n, x, \tilde{x}) = c^{-n} \int_{\longrightarrow} dy \frac{e^{-ixy}}{(-iy)^n} \int_{\longrightarrow} d\tilde{\tau} \frac{e^{-i\tilde{x} - \frac{ix}{c^2\tilde{\tau}}}}{(-i\tilde{\tau})^n}. \tag{D.2}$$

This is the correct branch choice, as is evident from putting $\tau = \tilde{\tau} = i$ and then following the branch continuously. We then use the Hankel representation of the Gamma function

$$\int_{\mathcal{H}} dt \, e^{-t}(-t)^{-z} = -\frac{2\pi i}{\Gamma(z)}, \tag{D.3}$$

Table 2: The first few values of $\tilde{m}(\frac{a}{c})$.

| $a$ \ $c$ | 0 | 1 | 2 | 3 | 4 | 5 | 6 | 7 | 8 | 9 | 10 | 11 | 12 | 13 | 14 | 15 |
|---|---|---|---|---|---|---|---|---|---|---|---|---|---|---|---|---|
| 1 | 0 | | | | | | | | | | | | | | | |
| 2 | | 0 | | | | | | | | | | | | | | |
| 3 | | 0 | 0 | | | | | | | | | | | | | |
| 4 | | 0 | | 1 | | | | | | | | | | | | |
| 5 | | 0 | 0 | 0 | 2 | | | | | | | | | | | |
| 6 | | 0 | | | | 3 | | | | | | | | | | |
| 7 | | 0 | 0 | 1 | 0 | 1 | 4 | | | | | | | | | |
| 8 | | 0 | | 0 | | 0 | | 5 | | | | | | | | |
| 9 | | 0 | 0 | | 2 | 0 | | 2 | 6 | | | | | | | |
| 10 | | 0 | | 1 | | | | 1 | | 7 | | | | | | |
| 11 | | 0 | 0 | 0 | 0 | 3 | 0 | 1 | 1 | 3 | 8 | | | | | |
| 12 | | 0 | | | | 1 | | 0 | | | | 9 | | | | |
| 13 | | 0 | 0 | 1 | 2 | 0 | 4 | 0 | 0 | 1 | 2 | 4 | 10 | | | |
| 14 | | 0 | | 0 | | 0 | | | | 2 | | 2 | | 11 | | |
| 15 | | 0 | 0 | | 0 | | | | 5 | 0 | | 2 | | 5 | 12 | |
| 16 | | 0 | | 1 | | 3 | | 2 | | 0 | | 1 | | 3 | | 13 |

where the Hankel contour $\mathcal{H}$ goes around the positive real axis in a counterclockwise sense. We can rotate the contour in the $y$ integral and deform it into the Hankel contour

$$\int_{\longrightarrow} \mathrm{d}y\, \frac{\mathrm{e}^{-ixy}}{(-iy)^n} = -ix^{n-1}\int_{\uparrow} \mathrm{d}t\, \mathrm{e}^{-t}(-t)^{-n} = ix^{n-1}\int_{\mathcal{H}} \mathrm{d}t\, \mathrm{e}^{-t}(-t)^{-n} = \frac{2\pi x^{n-1}}{\Gamma(n)}. \tag{D.4}$$

In the $\tilde{\tau}$ integral, we change variables to $\tilde{\tau} = \frac{i}{c}\sqrt{\frac{x}{\tilde{x}}}t$, which gives

$$\int_{\longrightarrow} \mathrm{d}\tilde{\tau}\, \frac{\mathrm{e}^{-i\tilde{x}\tilde{\tau}-\frac{ix}{c^2\tilde{\tau}}}}{(-i\tilde{\tau})^n} = -ic^{n-1}x^{-\frac{n-1}{2}}\tilde{x}^{\frac{n-1}{2}}\int_{\mathcal{H}_-} \mathrm{d}t\, \frac{\mathrm{e}^{\frac{\sqrt{x\tilde{x}}}{c}(t-t^{-1})}}{t^n}$$
$$= 2\pi c^{n-1}x^{-\frac{n-1}{2}}\tilde{x}^{\frac{n-1}{2}}J_{n-1}\left(\frac{2\sqrt{x\tilde{x}}}{c}\right), \tag{D.5}$$

where we have used the Schläfli's integral representation of the Bessel function [48, eq. (10.9.19)]. Here, $\mathcal{H}_-$ is the Hankel contour going around the negative real axis in a counterclockwise sense and we pick the principal branch of $t^{-n}$ on the intersection of the contour with the real axis.

Inserting these two evaluations in (D.2) yields

$$\mathcal{I}_c(n,x,\tilde{x}) = \frac{4\pi^2(x\tilde{x})^{\frac{n-1}{2}}J_{n-1}\left(\frac{2\sqrt{x\tilde{x}}}{c}\right)}{c\,\Gamma(n)}. \tag{D.6}$$

## D.2 Second integral

For the evaluation of the second integral (76) we notice that

$$\frac{\mathrm{e}^{-ix\tau-i\tilde{x}\tilde{\tau}}}{\left(\frac{1}{c}-c\tau\tilde{\tau}\right)^n}\left(\tau+\frac{1}{c^2\tilde{\tau}}\right) = \frac{i}{x}\left(\partial_\tau \mathrm{e}^{-ix\tau-i\tilde{x}\tilde{\tau}}\right)\frac{\tau+\frac{1}{c^2\tilde{\tau}}}{\left(\frac{1}{c}-c\tau\tilde{\tau}\right)^n}. \tag{D.7}$$

Therefore we can integrate $\tau$ by parts in the definition of $\mathcal{I}_c^{(2)}(n, x, \tilde{x})$ and obtain

$$
\begin{aligned}
\mathcal{I}_c^{(2)}(n, x, \tilde{x}) &= -\frac{i}{x} \int_{\longrightarrow} \mathrm{d}\tau \int_{\longrightarrow} \mathrm{d}\tilde{\tau}\, e^{-ix\tau - i\tilde{x}\tilde{\tau}} \partial_\tau \left( \frac{\tau + \frac{1}{c^2 \tilde{\tau}}}{\left(\frac{1}{c} - c\tau\tilde{\tau}\right)^n} \right) \\
&= \frac{i}{xc} \int_{\longrightarrow} \mathrm{d}\tau \int_{\longrightarrow} \mathrm{d}\tilde{\tau}\, e^{-ix\tau - i\tilde{x}\tilde{\tau}} \left( \frac{c(n-1)}{\left(\frac{1}{c} - c\tau\tilde{\tau}\right)^n} - \frac{2n}{\left(\frac{1}{c} - c\tau\tilde{\tau}\right)^{n+1}} \right) \\
&= \frac{i(n-1)}{x} \mathcal{I}_c^{(1)}(n, x, \tilde{x}) - \frac{2in}{xc} \mathcal{I}_c^{(1)}(n+1, x, \tilde{x}) \\
&= \frac{4\pi^2 i x^{\frac{n-3}{2}} \tilde{x}^{\frac{n-1}{2}}}{c\,\Gamma(n-1)} J_{n-1}\left(\frac{2\sqrt{x\tilde{x}}}{c}\right) - \frac{8\pi^2 i x^{\frac{n-2}{2}} \tilde{x}^{\frac{n}{2}}}{c^2\,\Gamma(n)} J_n\left(\frac{2\sqrt{x\tilde{x}}}{c}\right) \\
&= \frac{4\pi^2 i x^{\frac{n-2}{2}} \tilde{x}^{\frac{n}{2}}}{c^2\,\Gamma(n)} \left( J_{n-2}\left(\frac{2\sqrt{x\tilde{x}}}{c}\right) - J_n\left(\frac{2\sqrt{x\tilde{x}}}{c}\right) \right).
\end{aligned}
\tag{D.8}
$$

We used the recurrence relation of the Bessel function to pass to the last line.

# E   Useful modular transformations

In this appendix, we derive how the modular forms

$$
\frac{\vartheta_2(2\tau)}{\eta(\tau)^6}, \qquad \frac{\vartheta_3(2\tau)}{\eta(\tau)^6},
\tag{E.1}
$$

transform under an $\mathrm{SL}(2, \mathbb{Z})$ transformation. This was done in [49] by relating the numerators to eta-quotients. We shall follow another route, which is closely related to the origin of these functions as a string theoretic two-point function.

To start, notice that

$$
\frac{\vartheta_2(2\tau)}{\eta(\tau)^6} = \int_0^1 \mathrm{d}z\, \Phi(z, \tau),
\tag{E.2a}
$$

$$
\frac{\vartheta_3(2\tau)}{\eta(\tau)^6} = \int_0^1 \mathrm{d}z\, \frac{\vartheta_4(z, \tau)^2}{\eta(\tau)^6} = -\int_0^1 \mathrm{d}z\, e^{-2\pi i z - \frac{\pi i \tau}{2}} \Phi(z, \tau),
\tag{E.2b}
$$

and $\Phi(z, \tau) = \frac{\vartheta_1(z, \tau)^2}{\eta(\tau)^6}$ is a weak Jacobi form of weight $-2$ and index $1$, meaning that it has the following modular transformation behaviour,

$$
\Phi\left( \frac{z}{c\tau + d}, \frac{a\tau + b}{c\tau + d} \right) = (c\tau + d)^{-2} e^{\frac{2\pi i c z^2}{c\tau + d}} \Phi(z, \tau).
\tag{E.3}
$$

We also define the shifted Gauss sum:

$$
G_s(d, c) = \sum_{k \in \mathbb{Z}_c + \frac{s}{2}} e^{-\frac{2\pi i d k^2}{c}}.
\tag{E.4}
$$

These Gauss sums admit closed form expressions in terms of Jacobi symbols, which we will review in appendix E.3.

## E.1 $\quad \frac{\vartheta_2(2\tau)}{\eta(\tau)^6}$

We start by writing for $\tau' = \frac{a\tau+b}{c\tau+d}$:

$$
\begin{aligned}
\frac{\vartheta_2(2\tau')}{\eta(\tau')^6} &= \int_0^1 dz \, \Phi\left(z, \frac{a\tau+b}{c\tau+d}\right) \\
&= \frac{1}{(c\tau+d)^2} \int_0^1 dz \, e^{2\pi i c(c\tau+d)z^2} \Phi((c\tau+d)z, \tau) \\
&= \sum_{m,n\in\mathbb{Z}} \frac{(-1)^{m+n+1} q^{\frac{1}{2}\left(n+\frac{1}{2}\right)^2 + \frac{1}{2}\left(m+\frac{1}{2}\right)^2}}{(c\tau+d)^2 \eta(\tau)^6} \int_0^1 dz \, e^{2\pi i(c\tau+d)\left(cz^2+(n+m+1)z\right)} \, .
\end{aligned}
\tag{E.5}
$$

Let us assume $c > 0$ in the following, since the case with $c = 0$ is elementary. We now make the following change of variables in the sum: we set $k = \frac{m+n+1}{2}, \ell = \frac{m-n}{2}$, so that the sum over $m, n$ splits in two:

$$
\sum_{m,n\in\mathbb{Z}} (\cdots) = \left( \sum_{k\in\mathbb{Z}, \ell\in\mathbb{Z}+\frac{1}{2}} + \sum_{k\in\mathbb{Z}+\frac{1}{2}, \ell\in\mathbb{Z}} \right)(\cdots) \, .
\tag{E.6}
$$

When expressed through $k$ and $\ell$, the expression becomes

$$
\begin{aligned}
\frac{\vartheta_2(2\tau')}{\eta(\tau')^6} &= \left( \sum_{k\in\mathbb{Z}, \ell\in\mathbb{Z}+\frac{1}{2}} - \sum_{k\in\mathbb{Z}+\frac{1}{2}, \ell\in\mathbb{Z}} \right) \frac{q^{\ell^2} e^{-\frac{2\pi i d k^2}{c}}}{(c\tau+d)^2 \eta(\tau)^6} \int_0^1 dz \, e^{2\pi i c(c\tau+d)\left(z+\frac{k}{c}\right)^2} \\
&= \left( \vartheta_2(2\tau) \sum_{k\in\mathbb{Z}} - \vartheta_3(2\tau) \sum_{k\in\mathbb{Z}+\frac{1}{2}} \right) \frac{e^{-\frac{2\pi i d k^2}{c}}}{(c\tau+d)^2 \eta(\tau)^6} \int_0^1 dz \, e^{2\pi i c(c\tau+d)\left(z+\frac{k}{c}\right)^2} \, ,
\end{aligned}
\tag{E.7}
$$

where in the last step, we have performed the sums over $\ell$ and reconstructed $\vartheta_2(2\tau)$ and $\vartheta_3(2\tau)$. We now massage the sums over $k$. Let us write $k = cn + m$ with $n \in \mathbb{Z}$ and $m \in \{0, 1, \ldots, c-1\}$, so that for $s = 0, 1$,

$$
\begin{aligned}
\sum_{k\in\mathbb{Z}+\frac{s}{2}} e^{-\frac{2\pi i d k^2}{c}} \int_0^1 dz \, e^{2\pi i c(c\tau+d)\left(z+\frac{k}{c}\right)^2} &= \sum_{m\in\mathbb{Z}_c+\frac{s}{2}} \sum_{n\in\mathbb{Z}} e^{-\frac{2\pi i d m^2}{c}} \sum_{n\in\mathbb{Z}} \int_n^{n+1} dz \, e^{2\pi i c(c\tau+d)\left(z+\frac{\ell}{c}\right)^2} \\
&= G_s(d, c) \int_{-\infty}^{+\infty} dz \, e^{2\pi i c(c\tau+d)z^2} \\
&= G_s(d, c)(-2ic(c\tau+d))^{-\frac{1}{2}} \, .
\end{aligned}
\tag{E.8}
$$

We have used the fact that $\sum_{n\in\mathbb{Z}} \int_n^{n+1} dz \, f(z) = \int_{-\infty}^{\infty} dz \, f(z)$, and then shifted the integration variable $z \to z - \frac{m}{c}$. Putting everything together, we get the desired modular transformation:

$$
\frac{\vartheta_2(2\tau')}{\eta(\tau')^6} = \frac{1}{\sqrt{2c}\,(-i(c\tau+d))^{\frac{5}{2}}} \left[ -G_0(d, c)\frac{\vartheta_2(2\tau)}{\eta(\tau)^6} + G_1(d, c)\frac{\vartheta_3(2\tau)}{\eta(\tau)^6} \right] \, .
\tag{E.9}
$$

## E.2 $\quad \frac{\vartheta_3(2\tau)}{\eta(\tau)^6}$

We can employ the same logic to derive the modular transformation of $\frac{\vartheta_3(2\tau)}{\eta(\tau)^6}$. We have:

$$
\begin{aligned}
\frac{\vartheta_3(2\tau')}{\eta(\tau')^6} &= -\int_0^1 \mathrm{d}z\ e^{-2\pi i z - \frac{\pi i(a\tau+b)}{2(c\tau+d)}} \Phi\left(z, \frac{a\tau+b}{c\tau+d}\right) \\
&= -\frac{1}{(c\tau+d)^2} \int_0^1 \mathrm{d}z\ e^{-2\pi i z - \frac{\pi i(a\tau+b)}{2(c\tau+d)} + 2\pi i c(c\tau+d)z^2} \Phi(z, \tau) \\
&= \sum_{m,n\in\mathbb{Z}} \frac{(-1)^{m+n} q^{\frac{1}{2}\left(n+\frac{1}{2}\right)^2 + \frac{1}{2}\left(m+\frac{1}{2}\right)^2}}{(c\tau+d)^2 \eta(\tau)^6} \int_0^1 \mathrm{d}z\ e^{-2\pi i z - \frac{\pi i(a\tau+b)}{2(c\tau+d)} + 2\pi i(c\tau+d)(cz^2 + (m+n+1)z)} \\
&= \left(-\sum_{k\in\mathbb{Z}, \ell\in\mathbb{Z}+\frac{1}{2}} + \sum_{k\in\mathbb{Z}+\frac{1}{2}, \ell\in\mathbb{Z}}\right) \frac{q^{\ell^2} e^{-\frac{\pi i(4dk^2 - 4k+a)}{2c}}}{(c\tau+d)^2 \eta(\tau)^6} \int_0^1 \mathrm{d}z\ e^{2\pi i c(c\tau+d)\left(z - \frac{1}{2c(c\tau+d)} + \frac{k}{c}\right)^2} \\
&= \left(-\vartheta_2(2\tau)\sum_{k\in\mathbb{Z}} + \vartheta_3(2\tau)\sum_{k\in\mathbb{Z}+\frac{1}{2}}\right) \frac{e^{-\frac{\pi i(4dk^2 - 4k+a)}{2c}}}{(c\tau+d)^2 \eta(\tau)^6} \int_0^1 \mathrm{d}z\ e^{2\pi i c(c\tau+d)\left(z - \frac{1}{2c(c\tau+d)} + \frac{k}{c}\right)^2}.
\end{aligned}
$$

$$(\text{E.10})$$

We set as before $k = \frac{m+n+1}{2}, \ell = \frac{m-n}{2}$ and completed the square in the exponent. The sums over $k$ can again be evaluated by setting $k = cn + m$ with $m \in \mathbb{Z}_c$, which gives

$$
\frac{\vartheta_3(2\tau')}{\eta(\tau')^6} = \frac{1}{\sqrt{2c}(-i(c\tau+d))^{\frac{5}{2}}} \left( \frac{\vartheta_2(2\tau)}{\eta(\tau)^6} \sum_{m\in\mathbb{Z}_c} - \frac{\vartheta_3(2\tau)}{\eta(\tau)^6} \sum_{m\in\mathbb{Z}_c+\frac{1}{2}} \right) e^{-\frac{\pi i(4dm^2 - 4m+a)}{2c}}. \tag{E.11}
$$

We can complete the squares in the Gauss sums and relate them to the shifted Gauss sums that we introduced above (E.4),

$$
\sum_{m\in\mathbb{Z}_c+\frac{s}{2}} e^{-\frac{\pi i(4dm^2 - 4m+a)}{2c}} = \sum_{m\in\mathbb{Z}_c+\frac{s}{2}} e^{-\frac{2\pi id}{c}\left(m - \frac{a}{2}\right)^2 + \frac{\pi i(a-4m)(ad-1)}{2c}} = i^{(a+2s)b} G_{a+s}(d,c). \tag{E.12}
$$

Putting everything together then gives:

$$
\frac{\vartheta_3(2\tau')}{\eta(\tau')^6} = \frac{1}{\sqrt{2c}(-i(c\tau+d))^{\frac{5}{2}}} \left[ i^{ab} G_a(d,c) \frac{\vartheta_2(2\tau)}{\eta(\tau)^6} - i^{(a+2)b} G_{a+1}(d,c) \frac{\vartheta_3(2\tau)}{\eta(\tau)^6} \right]. \tag{E.13}
$$

This shows eq. (98).

### E.3 Evaluation of Gauss sums

We will now explain how to evaluate the Gauss sum

$$
G(d,c) = e^{\frac{\pi i}{c}} \sum_{m=0}^{c-1} e^{-\frac{2\pi i}{c}(dm^2 + (d-1)m)}, \tag{E.14}
$$

which is the sum that appears in the final answer. The result depends on $c \bmod 4$ in a crucial way.

### E.3.1 Odd $c$

For odd $c$, one can complete the square by shifting $m \to m - \frac{1}{2}a(d-1)(c+1)$, leading to

$$G(d,c) = -e^{\frac{\pi i \left(1-c^2\right)(a+d)}{2c}} \sum_{m \in \mathbb{Z}_c} e^{-\frac{2\pi i d m^2}{c}} = -\sqrt{c}\,\varepsilon(c)e^{\frac{\pi i \left(1-c^2\right)(a+d)}{2c}} \left(\frac{-d}{c}\right). \tag{E.15}$$

We used the standard evaluation of the Gauss sum when no linear term is present in terms of the Jacobi symbol. Here, $\varepsilon(c) = 1$ when $c \equiv 1 \bmod 4$ and $\varepsilon(c) = i$ when $c \equiv 3 \bmod 4$.

### E.3.2 Even $c$

When $c \equiv 2 \bmod 4$, the answer vanishes, since the term $m$ cancels with the term $m + \frac{c}{2}$. This leaves $c \equiv 0 \bmod 4$. In this case, $d$ is necessarily odd since $(c,d) = 1$. Since $d$ is odd, we can shift $m \to m + \frac{a(1-d)}{2}$ to complete the square as before. This leads to

$$G(d,c) = i^{-ab}e^{\frac{\pi i (a+d)}{2c}} \sum_{m \in \mathbb{Z}_c} e^{-\frac{2\pi i d m^2}{c}} = (1+i)\sqrt{c}\,\varepsilon(c-d)^{-1}i^{-ab}e^{\frac{\pi i (a+d)}{2c}} \left(\frac{c}{c-d}\right), \tag{E.16}$$

where we again used the standard evaluation of the quadratic Gauss sum for the case $c \equiv 0 \bmod 4$.

### E.3.3 Relation to Dedekind sums

We can relate the Jacobi symbol to Dedekind sums, which expresses this again in terms of more standard objects. The basic relation is that for $c$ odd and $(a,c) = 1$, we have

$$\left(\frac{a}{c}\right) = e^{\frac{\pi i}{4}(c-1-12c\,s(a,c))}. \tag{E.17}$$

Thus for odd $c$, we find with the help of $s(a,c) = s(a^*,c) = -s(-a,c)$

$$G(d,c) = -\sqrt{c}\,e^{\frac{\pi i (c^2-1)(c-4(a+d))}{8c}+3\pi i c\,s(a,c)}. \tag{E.18}$$

For $c \equiv 0 \bmod 4$, one uses the same relation together with the reciprocity of the Dedekind sum

$$s(a,c) + s(c,a) = \frac{1}{12}\left(\frac{a}{c} + \frac{c}{a} + \frac{1}{ac}\right) - \frac{1}{4}, \tag{E.19}$$

which brings the result into the form

$$G(d,c) = -\sqrt{2c}\,e^{-\frac{\pi i}{4c}\left(2a^2 d + d^2 - 4a - 2d + 1\right) - \frac{\pi i}{8}(d-1)^2 - 3\pi i(c-d)s(a,c)}. \tag{E.20}$$

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
