# Peer review of "Rademacher expansion of modular integrals"

_SciPost Physics, doi:SciPost Phys. 19, 103 (2025)_

## Round 2 · Referee Report · Anonymous (Referee 2) · 2025-6-27

Report
While such integrals are well-studied, they are typically hard to evaluate and new results are most welcome. Also in preparation to address higher loop amplitudes. The present paper puts forward a new method to evaluate such integrals, which is reminiscent of the contour integral for the Fourier coefficients of a weakly holomorphic modular form. I find the result novel, original and impressive. The authors moreover apply it to a variety of string amplitudes of interest.
The analysis is well-explained in the paper, which I think is very suitable for publication in SciPost.
Requested changes
I would suggest that the authors address a few general comments:
(1) I think the conditions on the integrand f for the stated results can be stated more clearly. My impression is that the authors analytically continue a function of two real variables (real and imaginary part of $\tau$) to a function of two complex variables $\tau$ and $\tilde \tau$. To justify the contour deformations, I think the integrand should be weakly holomorphic in both $\tau$ and $\tilde \tau$ separately, ie no singularities in the interior of $\mathbb{H}\times \mathbb{H}$, only potentially divergent when these variables approach the boundary.
(2) It would be helpful to express the analytically continued functions of $\tau$, $\tilde \tau$ explicitly in terms of $\tau$ and $\tilde \tau$. For example the rhs of (3.13) and rhs of (3.46).
(3) In the reduction to polar terms in Sec. 3.1, terms in the Fourier expansion are distinguished by whether $x, \tilde x<$ or $>0$, while terms with $x,\tilde x=0$ do not seem commented on. I believe these terms may be important and once understood may clarify the issue of section 3.6.2.
Recall that in the holomorphic setting for the constant term of a weight 0 modular form, the error terms in restricting to polar terms do not vanish. The Circle Method is therefore strictly speaking not applicable for that coefficient.
A few minor comments are:
(1) p. 7 below (1.3): suggest to specify that this is the Bessel function of the first kind
(2) p. 29, Eq. (3.19)/(3.20) and p. 32/33, Eq. (3.37)/(3.38): I believe the first digits were first given by Marcus (1989), Eq. (19) and Eq. (18) respectively, in a slightly different normalisation. A reference seems appropriate.
Recommendation
Publish (surpasses expectations and criteria for this Journal; among top 10%)

---

## Round 2 · Referee Report · Anonymous (Referee 1) · 2025-6-27

Report
The results are successfully checked against numerical evaluations of the modular integral, and provide rapidly convergent infinite sum representations with fast convergence properties, so long as the integrand satisfies suitable growth conditions near the cusp which are carefully spelled out.
While I have not been able to check all the steps in the derivation, I find that this paper is very original, well written, and that the results are very striking, with potentially many applications, both in string theory and beyond. I strongly recommend publication in SciPost Physics, after the authors consider the following minor suggestions:
Requested changes
-
The pillow coordinate map is mentioned twice in the main body of the paper, but one has to dig into section C.1 to see it explicitly written out. I suggest referring to eq C.2 for example in the paragraph after eq (2.15).
-
The 'reciprocity relation' of the Dedekind sum is mentioned twice, but not stated anywhere, as far as I could see.
-
In eq. (3.2), along with the modular transformation of $g$, it would be good to specify that of $f$
-
The closed form result in (3.20) for $\Im Z$ in the bosonic string was obtained in Eq 3.40 of Manschot Wang [3], while another convergent representation of $\Re Z$ was obtained in Eq 3.14 of the same paper (up to overall factor of 2). A reference would be appropriate.
-
The phrase "Direct integration is clearly subpar" on p30 is subpar compared to the perfect style in the rest of the paper !
-
At the bottom of p33, the sentence "the four entries of $\rho_{a,b,d}^{k,l}$ have $(3-k)(3-l)$ polar terms is confusing, since the entries of $\rho$ are just real numbers. The sentence probably refers to the number of non-zero entries in $\rho_{a,b,d}^{k,l} M^k(x) \tilde M^l(x)$ in Eq (3.5).
-
In the discussion below 3.64 and 3.73, I would suggest to use Richardson transform to accelerate convergence.
-
In the paragraph below 3.65, the condition $mp+wq=1$ should be $mq+wp=1$.
-
Typo above eq 2.1: careful -> carefully
-
Typo below eq 2.60: cancels -> cancel
Recommendation
Publish (surpasses expectations and criteria for this Journal; among top 10%)

---

## Editorial Decision

published